# Dynamic gene regulation by nuclear colony-stimulating factor 1 receptor in human monocytes and macrophages

Laura Bencheikh[1,2], M'Boyba Khadija Diop [1], Julie Rivière[1], Aygun Imanci[1,2], Gerard Pierron[3], Sylvie Souquere[3], Audrey Naimo[4], Margot Morabito[1], Michaël Dussiot[5,6,7], Frédéric De Leeuw[8], Camille Lobry [1,2], Eric Solary[1,2,9,10] & Nathalie Droin [1,2,4,10]

Despite their location at the cell surface, several receptor tyrosine kinases (RTK) are also found in the nucleus, as either intracellular domains or full length proteins. However, their potential nuclear functions remain poorly understood. Here we find that a fraction of full length Colony Stimulating Factor-1 Receptor (CSF-1R), an RTK involved in monocyte/macrophage generation, migrates to the nucleus upon CSF-1 stimulation in human primary monocytes. Chromatin-immunoprecipitation identifies the preferential recruitment of CSF-1R to intergenic regions, where it co-localizes with H3K4me1 and interacts with the transcription factor EGR1. When monocytes are differentiated into macrophages with CSF-1, CSF-1R is redirected to transcription starting sites, colocalizes with H3K4me3, and interacts with ELK and YY1 transcription factors. CSF-1R expression and chromatin recruitment is modulated by small molecule CSF-1R inhibitors and altered in monocytes from chronic myelomonocytic leukemia patients. Unraveling this dynamic non-canonical CSF-1R function suggests new avenues to explore the poorly understood functions of this receptor and its ligands.

[1] INSERM U1170, Gustave Roussy Cancer Center, 94805 Villejuif, France. [2] Faculté de Médecine, Université Paris-Sud, 94270 Le Kremlin-Bicêtre, France. [3] CNRS UMR9196, Gustave Roussy Cancer Center, 94805 Villejuif, France. [4] INSERM US23, CNRS UMS 3655, AMMICa, Genomic platform, Gustave Roussy Cancer Center, 94805 Villejuif, France. [5] INSERM U1163, CNRS UMR8254, Institut Imagine, Hôpital Necker Enfants Malades, 75015 Paris, France. [6] Institut Imagine, Hôpital Necker Enfants Malades, Université Sorbonne-Paris-Cité, 75015 Paris, France. [7] Laboratoire d'excellence GR-Ex, Institut Imagine, Hôpital Necker Enfants Malades, 75015 Paris, France. [8] INSERM US23, CNRS UMS 3655, AMMICa, Imaging and Cytometry Platform, Gustave Roussy Cancer Center, 94805 Villejuif, France. [9] Department of Hematology, Gustave Roussy Cancer Center, 94805 Villejuif, France. [10] These authors contributed equally: Eric Solary, Nathalie Droin. Correspondence and requests for materials should be addressed to E.S. (email: eric.solary@gustaveroussy.fr) or to N.D. (email: nathalie.droin@gustaveroussy.fr)

Receptor tyrosine kinases (RTK) constitute the largest family of catalytic receptors with 58 members[1]. Upon stimulation, RTKs undergo homo-oligomers or hetero-oligomers formation required for cytoplasmic catalytic domain auto-phosphorylation. In turn, multiple signaling pathways are activated and relay information from the cell membrane to intracellular compartments to trigger critical cellular processes[1]. Beside this canonical mode of signaling, several RTKs are also present in the cell nucleus as either intracellular domains (ICD), or full length proteins[2,3]. ICD are generated by proteolytic cleavage or alternative mRNA splicing[3]. Full length RTKs usually translocate from the cell surface to the nucleus through Golgi apparatus and endoplasmic reticulum[4] before nuclear addressing through importin β-mediated pathways[5]. In the nucleus, RTKs phosphorylate proteins[6,7] and regulate gene transcription[6] to modulate DNA damage response[7], cell proliferation[6], survival[8], and migration[9]. Therapeutic strategies targeting RTKs were developed, mostly to treat cancers, and RTKs nuclear accumulation could modulate their efficacy[10].

Colony-Stimulating Factor 1 Receptor (CSF-1R), also called M-CSFR (Macrophage colony-stimulating factor), is encoded by CFMS (Cellular Feline McDonough Sarcoma) proto oncogene and is a class III RTK expressed by phagocytic mononuclear cells[11]. CSF-1R is involved in several human diseases, its constitutive inactivating mutation induces a leuko-encephalopathy while its stimulation supports tumor progression and chronic inflammatory diseases. CSF-1R-targeting strategies are currently tested clinically in treating tumor infiltrated with macrophages[12].

Upon binding of either Colony-Stimulating Factor 1 (CSF-1) or interleukin (IL)-34[13], its two known ligands, CSF-1R undergoes oligomerization, relieving catalytic domain inhibition[14] and activating signal transduction[15,16], which is subsequently attenuated by CSF-1R ubiquitinylation, internalization, and degradation[17]. The nuclear localization of CSF-1R remains a controversial issue. CSF-1R proteolytic cleavage generating an ICD that migrates in the nucleus was initially reported[18]. Holoreceptor nuclear localization was subsequently described in human epithelial cancer cell lines[19] and murine macrophages[20], where CSF-1R could be phosphorylated[20] and binds the promoter of selected genes[19]. Little is known, however about nuclear CSF-1R functional importance in physiological and pathological conditions.

Here, we demonstrate that a fraction of CSF-1R is in the nucleus of human monocytes, where it is recruited to DNA, co-localizes with H3K4me1 histone marks and interacts with EGR1. Upon CSF-1 exposure, which induces monocyte differentiation into macrophages, CSF-1R localization on chromatin changes within a few hours, it colocalizes with H3K4me3, and promotes gene expression through interaction with transcription factors YY1 and ELK. This function is affected by small molecule CSF-1R inhibitors and altered in dysplastic monocytes collected from chronic myelomonocytic leukemia (CMML) patients. These results emphasize a dynamic role of nuclear CSF-1R in monocyte differentiation into macrophages.

## Results

### A fraction of CSF-1R is located in human monocyte nucleus.
We sorted human monocytes from healthy donor peripheral blood and detected CSF-1R in both the cytoplasm and the nucleus by confocal microscopy (Fig. 1a, b), which was further confirmed by orthogonal views (Fig. 1c). The signal specificity was confirmed by the use of a blocking peptide mimicking the epitope recognized by CSF-1R antibody, and by monocyte transfection with CFMS siRNA (Fig. 1d). CSF-1R was also detected in monocyte nucleus by imaging flow cytometry

(Supplementary Fig. 1a, b). Monocyte fractionation into nuclear versus cytoplasmic and membrane fractions followed by immunoblotting confirmed CSF-1R nuclear detection as a full-length protein with partially (130 kDa) and fully glycosylated (170 kDa) forms (Fig. 1e). Again, signal specificity was demonstrated by two CFMS targeting siRNAs, which totally abolished the signal in the nucleus and the cytoplasmic and membrane fractions (Supplementary Fig. 1c). Finally, CSF-1R localization was observed by electron microscopy in heterochromatin and euchromatin (Fig. 1f), which was validated by a distinct antibody targeting CSF-1R N-terminal fragment (Supplementary Fig. 1d). Monocyte fractionation without denaturation, followed by immunoblotting, detected a transient CSF-1R dimerization in the membrane and cytoplasmic fraction after 10 min of CSF-1 treatment, which was not detected in nuclear extracts, even after prolonged immunoblot exposure, suggesting the nuclear expression of monomeric holoreceptor (Supplementary Fig. 1e). All together, these results demonstrate the presence of a fraction of full-length CSF-1R in human monocyte nucleus.

### Nuclear CSF-1R holoreceptor location is driven by its ligand.
Peripheral blood monocytes were exposed to AF488-labeled recombinant CSF-1 for 15 min before fixation and staining with anti-CSF-1R antibody. As expected, CSF-1 and CSF-1R co-localized mainly at the plasma membrane and in the cytoplasm. CSF-1 was also detected in monocyte nucleus where it co-localizes with CSF-1R (Fig. 2a and movie as Supplementary Movie 1). Of note, CSF-1R staining after CSF-1 treatment (Fig. 2a) was more punctual compared with resting monocytes (Fig. 1a). This nuclear localization of CSF-1 and CSF-1R could not be related to nuclear localization signals (NLS) as CSF-1R primary sequence is devoid of this sequence and the putative NLS (amino acids 521 to 524) in CSF-1 sequence[19] is deleted from the recombinant CSF-1 used in this experiment. CSF-1 nuclear accumulation could be prevented by monocyte pre-treatment for 3 h with small molecule CSF-1R inhibitors, either BLZ945 or GW2580 (Fig. 2b). Confocal imaging further showed that monocyte exposure to CSF-1R inhibitors for 3 h partially depleted nuclear CSF-1R (Fig. 2c), which could be prevented by leptomycin B, an inhibitor of CRM1-mediated nuclear export (Fig. 2d). All together, these results suggest a role for CSF-1 in CSF-1R nuclear localization in human monocytes.

### CSF-1R is recruited on chromatin in human monocyte nucleus.
Chromatin immunoprecipitation followed by deep sequencing (ChIP-Seq) was performed with an anti-CSF-1R antibody in monocytes sorted from three healthy donors. Broad peak calling using MACS2 algorithm identified a mean number of 36,884 peaks, of which 4980 (13%) were common to the three samples (Fig. 3a and Supplementary Data 1). Intersection with Irreproducible Discovery Rate (IDR) analysis selected a set of 2303 common peaks (Supplementary Data 2). Cis-Regulatory Annotation System (CEAS) indicated that CSF-1R was preferentially recruited to intergenic (63.5%) regions (Fig. 3b). To gain insight into CSF-1R function at the chromatin level, we performed genome-wide analysis of H3K4me1 and H3K4me3 marks by ChIP-seq. For example, CSF-1R was recruited upstream of KLF6, PU.1 (Fig. 3c, d), CSF2RB and CEBPD genes (Supplementary Fig. 2a) and to the last intron of PU.1 gene (Fig. 3d). CSF-1R co-localization with histone mark H3K4me1 suggests regulating/enhancer regions[21,22]. ChIP-seq results were validated by ChIP-qPCR in independent healthy donor monocytes with two anti-CSF-1R antibodies that recognize its N-terminal and C-terminal parts, respectively (Supplementary Fig. 2b). Motif analysis of ChIP-seq data using HOMER, focused on peaks shared by the three donors, indicated that CSF-1R could be recruited on several

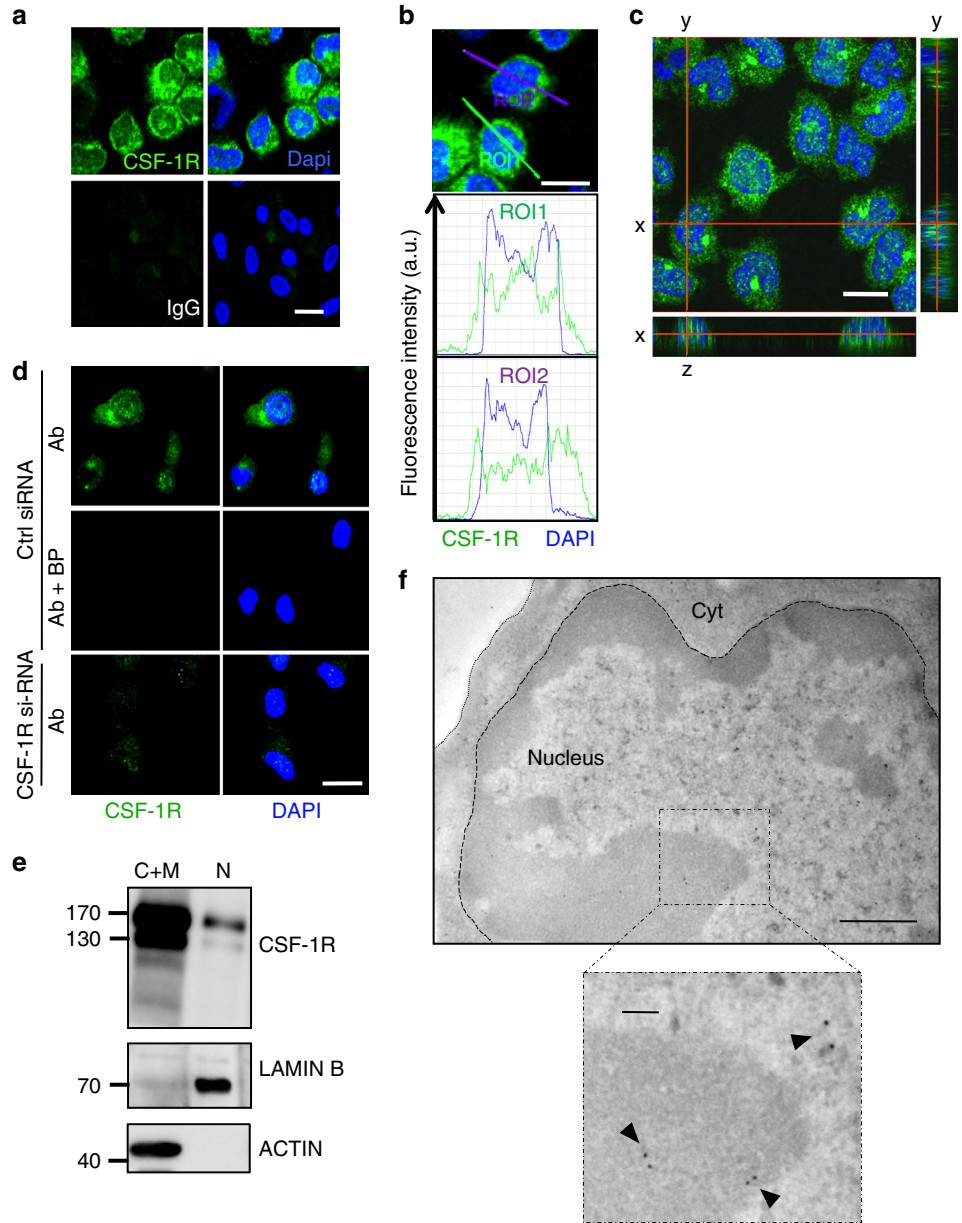

**Fig. 1** A fraction of CSF-1R is located in the nucleus of human monocytes. **a** Sorted peripheral blood human monocytes were stained with an anti-CSF-1R antibody (Cter sc-692) or a control IgG (green) and Dapi (blue), followed by confocal imaging analysis ($n = 3$, scale: 10 μm). **b** Quantification of the signal generated by CSF-1R labeling (green) and Dapi (blue) according to indicated axes (A.U.: arbitrary units, ROI: region of interest, scale: 10 μm). **c** Monocytes were stained with an anti-CSF-1R antibody (Cter sc-692) and Dapi (blue), followed by confocal imaging analysis (stack of 50 pictures of 0.2 μm) to reconstitute an orthogonal view (scale: 10 μm). **d** The specificity of CSF-1R labeling with sc-692 antibody (Ab in green) was explored in monocytes transfected 24 h before with a CSF-1R specific or a control si-RNA and treated or not with sc-692 blocking peptide (BP, $n = 2$, scale: 10 μm, Dapi in blue). **e** Monocytes were fractionated into cytoplasmic plus membrane (C+M) and nuclear (N) fractions and analyzed by immunoblotting with antibodies that recognize CSF-1R (Cell signaling #3152), Lamin B (N fraction) and actin (C+M fraction) ($n = 3$). **f** Immunogold analysis of CSF-1R expression in monocytes using the anti-Cter sc-692 antibody and electron microscopy ($n = 2$, scale: 500 nm; insert: 100 nm)

transcription factor binding sites, the most significant being EGR2 and EGR1 motifs (Fig. 3e). The ten biological pathways with highest enrichment identified by gene ontology (GO) analysis were related to monocyte and macrophage functions (Supplementary Table 1), suggesting a role for nuclear CSF-1R in supporting monocyte trophic functions.

**CSF-1R interacts with EGR1 to downregulate gene expression.** Since CSF-1R is recruited on EGR1 motifs and EGR1 is a transcription factor involved in monocytopoiesis[23], we performed co-

immunoprecipitation experiments in whole monocyte extracts, which demonstrated an interaction between the two proteins (Fig. 4a). We reasoned that CSF-1R and EGR1 peaks might colocalize in genome-wide analysis. ChIP-seq experiments were performed on two healthy donor monocyte samples with an anti-EGR1 antibody. Broad peak calling using MACS2 algorithm identified 3542 common peaks (Supplementary Data 3). A large fraction of EGR1 peaks (56.9%) co-localized with CSF-1R peaks and were mainly localized to intergenic regions (Fig. 4b, Supplementary Fig. 3a and Supplementary Data 4). Ranking heatmap centered on CSF-1R common peaks further showed their

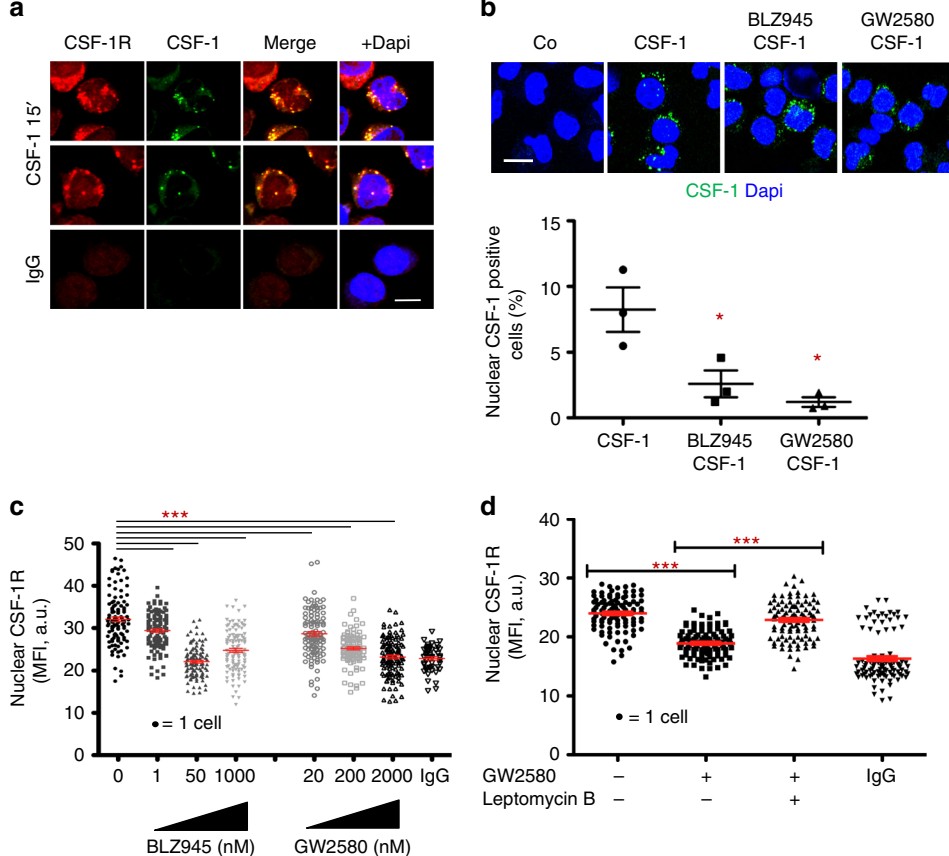

**Fig. 2** Nuclear localization of CSF-1R holoreceptor is driven by its ligand. **a** Peripheral blood monocytes were fixed and stained for CSF-1R or control IgG (red) and Dapi (blue) after 15 min exposure to AF488-labeled CSF-1 (1 μg/mL in green) followed by confocal imaging analysis ($n = 3$, scale: 7 μm). **b** Monocytes were pre-treated for 3 h with CSF-1R small molecule inhibitors (BLZ945 50 nM; GW2580 200 nM or control DMSO), exposed 15 min to AF488-labeled CSF-1 (except Co, 1 μg/mL, in green), fixed and stained for Dapi (blue) before analysis by confocal imaging; Lower panel: quantification of nuclear CSF-1 positive cells (mean +/− SEM of 3 independent experiments, scale: 10 μm, Mann–Whitney test: *$P < 0.05$). **c** Monocytes were treated or not with indicated concentrations of BLZ945 and GW2580 before imaging flow cytometry analysis of CSF-1R nuclear localization (MFI, mean fluorescence intensity, mean +/− SEM, Unpaired $t$ test: ***$P < 0.001$, $n = 3$). **d** Monocytes were treated or not for 3 h with 200 nM GW2580, in the absence or presence of 20 nM Leptomycin B, fixed and stained for CSF-1R or control IgG before imaging flow cytometry analysis of CSF-1R nuclear localization (mean +/− SEM, Paired $t$ test: ***$P < 0.001$, $n = 2$)

colocalisation with EGR1 and H3K4me1 while being distant from H3K4me3 peaks (Fig. 4c). We also observed that 78.2% of common CSF-1R and EGR1 peaks colocalized with H3K4me1 mark (Supplementary Data 5), which is exemplified on specific genes, namely *PU.1*, *KLF6* (Fig. 4d), *C3*, *CEBPD* and *CMKLR1* (Supplementary Fig. 3b). Motif analysis obviously identified EGR2 and EGR1 motifs as EGR1 recruitment sequences (Supplementary Fig. 3c). Collectively, these results argue for EGR1 and CSF-1R co-recruitment on chromatin with H3K4me1 mark in monocyte nucleus.

To explore the role of EGR1 in recruiting CSF-1R at the chromatin level, we deleted *EGR1* gene in THP1 monocytic cell line using Crisper/Cas9 technology. Preliminary experiments detected 2124 CSF-1R peaks common to primary human monocytes and THP1 cells. We obtained three clones with *EGR1* homozygous deletion, which was validated by Sanger sequencing (Supplementary Fig. 4a) and RT-qPCR (Supplementary Fig. 4b). We performed CSF-1R ChIP-seq analysis in these clones and their wildtype counterpart. Importantly, the quantity of DNA captured by the anti-CSF-1R antibody in EGR1-deleted clones was very low as compared with wild type cells (Supplementary Fig. 4c). Deep sequencing of these libraries, whose results were pooled for analysis, indicated that EGR1

deletion abrogated CSF-1R localization at EGR1 sites, for example on *PU.1*, *CEBPD* (Fig. 4e), *CALML5, TLR10, TOM1*, and *ROR2* genes (Supplementary Fig. 4d).

We then transfected monocytes with *CFMS*-targeting or scrambled siRNA to establish whether CSF-1R could be involved in gene regulation. CSF-1R downregulation induced a significant increase in the expression of several genes on which CSF1R was detected by ChIP-seq experiments, including *CEBPD, SRC, FGR*, and *KLF6* (Fig. 5a). Since this experiment could not uncouple membrane and nuclear CSF-1R functions, we also cloned four identified EGR1 motif sequences upstream of a *LUCIFERASE* reporter gene and transfected monocytes with this construct and *CFMS*-specific or control siRNA. An increased *LUCIFERASE* expression was observed after CSF-1R downregulation, suggesting a repressive role of CSF-1R on EGR1 sites (Fig. 5b). Since ChIP-seq experiments identified CSF-1R and EGR1 on *PU.1* promoter (Fig. 4d), we also cloned *PU.1* promoter with (PU.1 promoter 1) or without (PU.1 promoter 2) CSF-1R recruitment site upstream of *LUCIFERASE* gene. In monocytes transfected with the first plasmid, we observed an increased *LUCIFERASE* expression after CSF-1R downregulation (Fig. 5c), together with an increased expression of *PU.1* (Fig. 5d). These effects were not observed with the second plasmid encoding a promoter deleted of CSF-1R

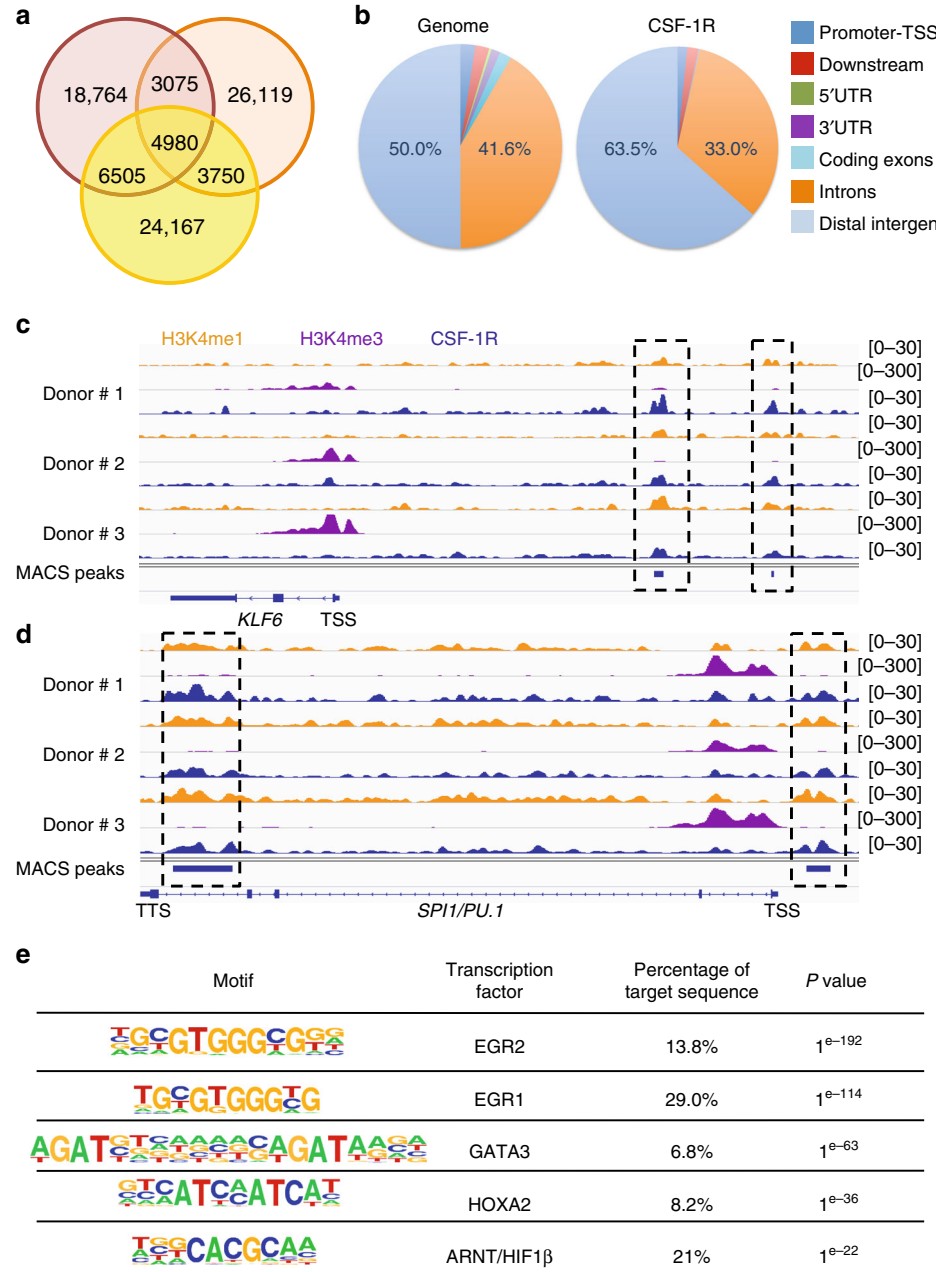

**Fig. 3** CSF-1R is recruited to the chromatin in human monocyte nucleus. **a** Venn diagram of peak calling (input normalization) from ChIP-seq experiment with sc-46662 anti-CSF-1R antibody performed on peripheral blood monocytes collected from 3 healthy donors. **b** Repartition of the ChIP-seq peaks common to the 3 donors on the genome (right panel) as compared with hg19 reference genome annotation (left panel). **c**, **d** Peak calling for CSF-1R (blue), H3K4me1 (orange) and H3K4me3 (purple). ChIP-seq on *KLF6* (**c**) and *SPI1/PU.1* (**d**) genes in the three healthy donor monocytes (TSS: transcription starting site, TTS: transcription termination site). **e** Motif analysis of ChIP-seq peaks common to the 3 healthy donor monocyte samples

recruitment site (Fig. 5c, d). Collectively, these data indicate that CSF-1R negatively regulates *PU.1* gene expression in human monocytes.

**Differential CSF-1R chromatin localization in macrophages.** Primary monocytes exposed to CSF-1 during 72 h differentiate into macrophages. Immunoblot analysis of nuclear extracts collected during this process detected a transient decrease in nuclear CSF-1R expression, 6 to 24 h after CSF-1 stimulation. After 72 h, the lower molecular weight CSF-1R was increased in macrophages (Fig. 6a). CSF-1R nuclear staining was quantified over the time after CSF-1 treatment by confocal imaging, showing again a

decrease of CSF-1R nuclear localization after 24 and 72 h compared with resting monocytes (Fig. 6b). Confocal imaging also detected CSF-1R in the nucleus of a fraction of these macrophages (Fig. 6c, d), which was confirmed by orthogonal views (Fig. 6e) and further validated by immunoglold staining and electron microscopy (Fig. 6f). Interestingly, nuclear CSF-1R was mainly located on the verge of heterochromatin, which suggested its association with actively transcribed regions[24].

CSF-1R ChIP-Seq was performed in macrophages generated by exposure of monocytes from 3 healthy donors to CSF-1 during 72 h (same donors used in Fig. 3). Peak calling identified a mean number of 29,060 peaks, of which 19.3% to the three samples (Supplementary Fig. 5a). Common peaks (Supplementary Data 6)

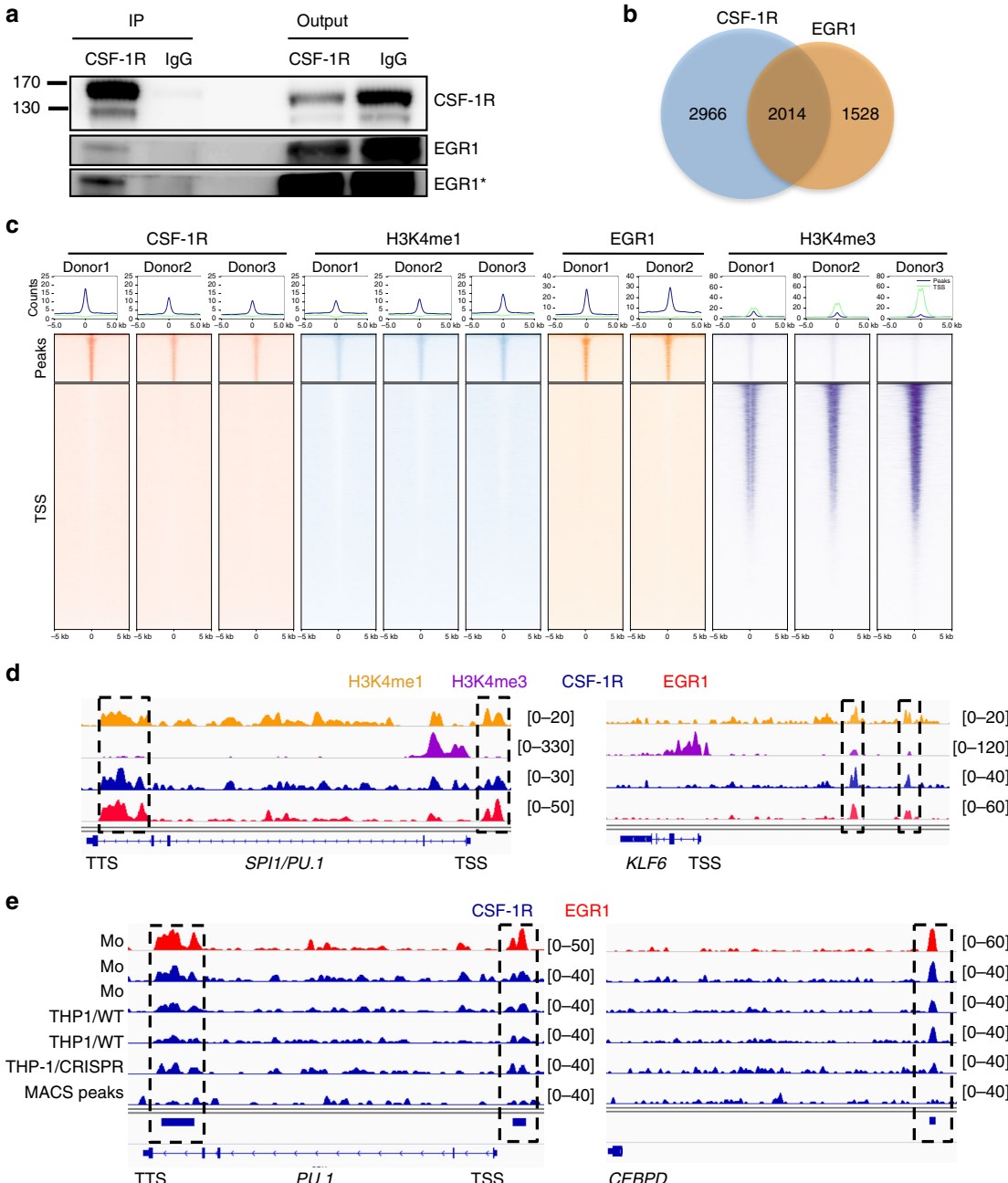

**Fig. 4** CSF-1R interacts with EGR1 to modulate gene expression. **a** Co-immunoprecipitation experiment using an anti-CSF-1R (sc-365719) or a control IgG in monocytes followed by immunoblotting with anti-CSF-1R and anti-EGR1 antibodies (n = 3). Asterisk (*) indicates a longer exposure of the blot. **b** Venn diagram of peak calling (input normalization) common to three donors from ChIP-seq experiment with sc-46662 anti-CSF-1R and sc-110X anti-EGR1 antibodies. **c** Ranking heatmap centered on CSF-1R peaks (top) and on gene transcription starting sites (TSS) of CSF-1R, H3K4me1, EGR1 and H3K4me3 peaks in monocytes. **d** Peak calling for H3K4me1 (orange), H3K4me3 (purple), EGR1 (red), CSF-1R (blue). ChIP-seq on *PU.1* and *KLF6* genes in monocytes (TSS: transcription starting site, TTS: transcription termination site). **e** Peak calling on *PU.1* and *CEBPD* genes for EGR1 in monocytes and for CSF-1R in monocytes (Mo) from two donors (in blue, same donors as in Fig. 3), wild-type THP-1 clones (THP1/WT), THP-1 clones deleted for EGR1 (pool of 3 clones, THP1/CRISPR)

indicated that CSF-1R was less recruited to intergenic regions (26.4%) as compared with untreated monocytes (63.5%, Fig. 3b) while a significant fraction of CSF-1R was detected on promoter-TSS (transcription starting site) (11.9%), exonic (9.1%) and 5′ UTR (9.9%) regions (Supplementary Fig. 5b). CSF-1R promoter-TSS recruitment was validated by ChIP-qPCR for *CFMS*, *MYB*, *EZH2*, *FOS*, *YY1* and *IL6R* genes in macrophages from independent healthy donors, *RAC2* being used as a negative control (Supplementary Fig. 5c).

Comparison of peak localization in monocytes and macrophages identified 1192 common peaks (representing 23.9% and

21.2% of monocyte and macrophage peaks, respectively, Supplementary Data 7). ChIP-Seq experiments with histone marks H3K4me1 and H3K4me3 performed on the same samples showed a higher fraction of CSF-1R associated with H3K4me1 in monocytes and H3K4me3 in macrophages (Fig. 7a and Supplementary Data 8–11, respectively). Ranking heatmaps centered on TSS indicated that CSF-1R is relocalized around the TSS where it colocalizes with H3K4me3 in macrophages (Fig. 7b), as examplified for *MAFB*, *JUN,* and *MYC* genes (Supplementary Fig. 5d). CSF-1R location also changed from the promoter region of *PU.1* gene in monocytes to *PU.1* first intron in

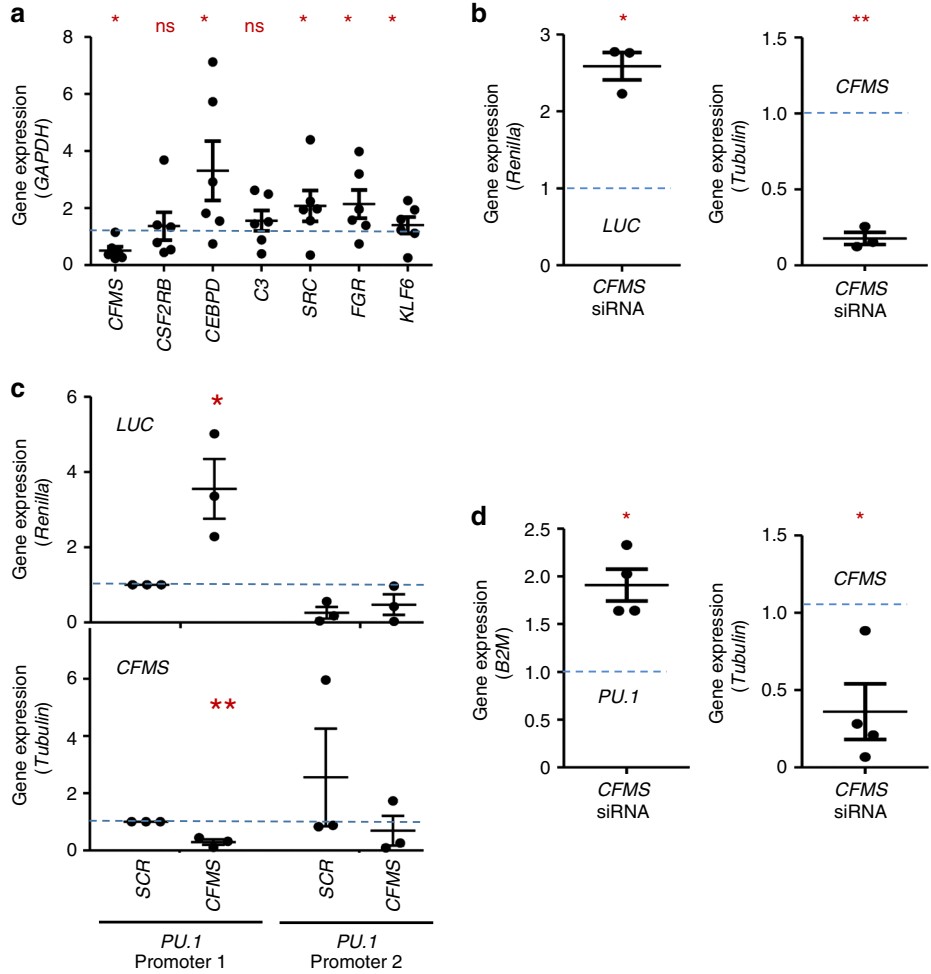

**Fig. 5** CSF-1R negatively regulates *PU.1* gene expression in human monocytes. **a** The expression of indicated genes was measured by RT-qPCR in monocytes transfected 72 h before with a scramble (*SCR*) or a *CFMS* specific siRNA (mean +/− SEM of 6 independent experiments, results in *CFMS*-siRNA transfected cells normalized to *SCR*-siRNA = blue hatched line; Mann–Whitney test: *$P < 0.05$; ns, non significant). **b** Expression of *LUCIFERASE* (*LUC*, normalized to *Renilla*) and *CFMS* (normalized to *TUBULIN*) genes in monocytes, 48 h after transfection with a scramble (*SCR*) or a *CFMS* specific siRNA and PGL4.17-EGR1 motif plasmid (mean +/− SEM of 3 independent experiments, results in *CFMS*-siRNA transfected cells normalized to *SCR*-siRNA = blue hatched line; Paired *t* test: *$P < 0.05$, **$P < 0.01$). **c** Expression of *LUCIFERASE* (normalized to *Renilla*) and *CFMS* (normalized to *TUBULIN*) in monocytes 48 h after transfection with *SCR* or *CFMS* specific siRNA and PGL4.17-PU1 promoter 1 (complete) or PGL4.17-PU1 promoter 2 (deleted of CSF-1R recruitment site) plasmids (mean +/− SEM of 3 independent experiments, normalization to control siRNA with *PU.1* promoter 1 = blue hatched line, Paired *t* test: *$P < 0.05$, **$P < 0.01$). **d** Expression of *PU.1* and *CFMS* genes in monocytes 72 h after transfection with *SCR* or *CFMS* specific siRNA (mean +/− SEM of 4 independent experiments, normalization to control siRNA, Paired *t* test: *$P < 0.05$)

macrophages where it colocalizes with H3K4me3 mark (Supplementary Fig. 5e). GO analysis of CSF-1R-interacting genes in macrophages showed a dramatic change as compared with monocytes as biological pathways with highest enrichment were related mainly to epigenetics and transcription regulation (Supplementary Table 2). Enforcing this dynamic process, motif analysis revealed CSF-1R recruitment on different transcription factor binding sites such as E2F4, ELK4, YY1 (yin yang 1), and ELK1, not seen in monocytes (Fig. 7c). All together, CSF-1-induced monocyte differentiation into macrophages modifies CSF-1R recruitment on chromatin. ELK proteins bind to serum response elements to modulate immune response[25,26], while YY1 is involved in macrophage response to lipopolysaccharides[27]. Co-immunoprecipitation (Fig. 7d) and immunofluorescence (Supplementary Fig. 5f) experiments in macrophages argued for CSF-1R interaction with ELK1 and YY1 proteins.

siRNA mediated downregulation of CSF-1R in CSF-1 treated monocytes induced a significant decrease in *YY1*, *ASXL1*, *CBL*, and *CJUN* expression (Supplementary Fig. 6a), supporting the

idea that CSF-1R may directly or indirectly participate to the transcription of these genes in macrophages. We then selected, two genes whose expression upon CSF-1R downregulation was compared in monocytes and macrophages. The first one is *KLF6* whose expression increases in monocytes when CSF-1R is downregulated (Fig. 5a) while being not affected by CSF-1R downregulation in macrophages (Supplementary Fig. 6b). The other one is *CSF2RB*, which is not significantly modified by *CFMS* siRNA in monocytes (Fig. 5a) while its expression is dramatically decreased by CSF-1R downregulation in macrophages (Supplementary Fig. 6b).

**CSF-1 modulates fastly CSF-1R recruitment on chromatin.** Having observed that monocyte exposure to CSF-1 for 72 h induced CSF-1R relocalization on gene TSS, we explored the kinetics of this event by treating monocytes with CSF-1 for only 6 h. ChIP-seq experiments (Supplementary Data 12) identified an increase in CSF-1R recruitment on exonic, promoter-TSS, TTS

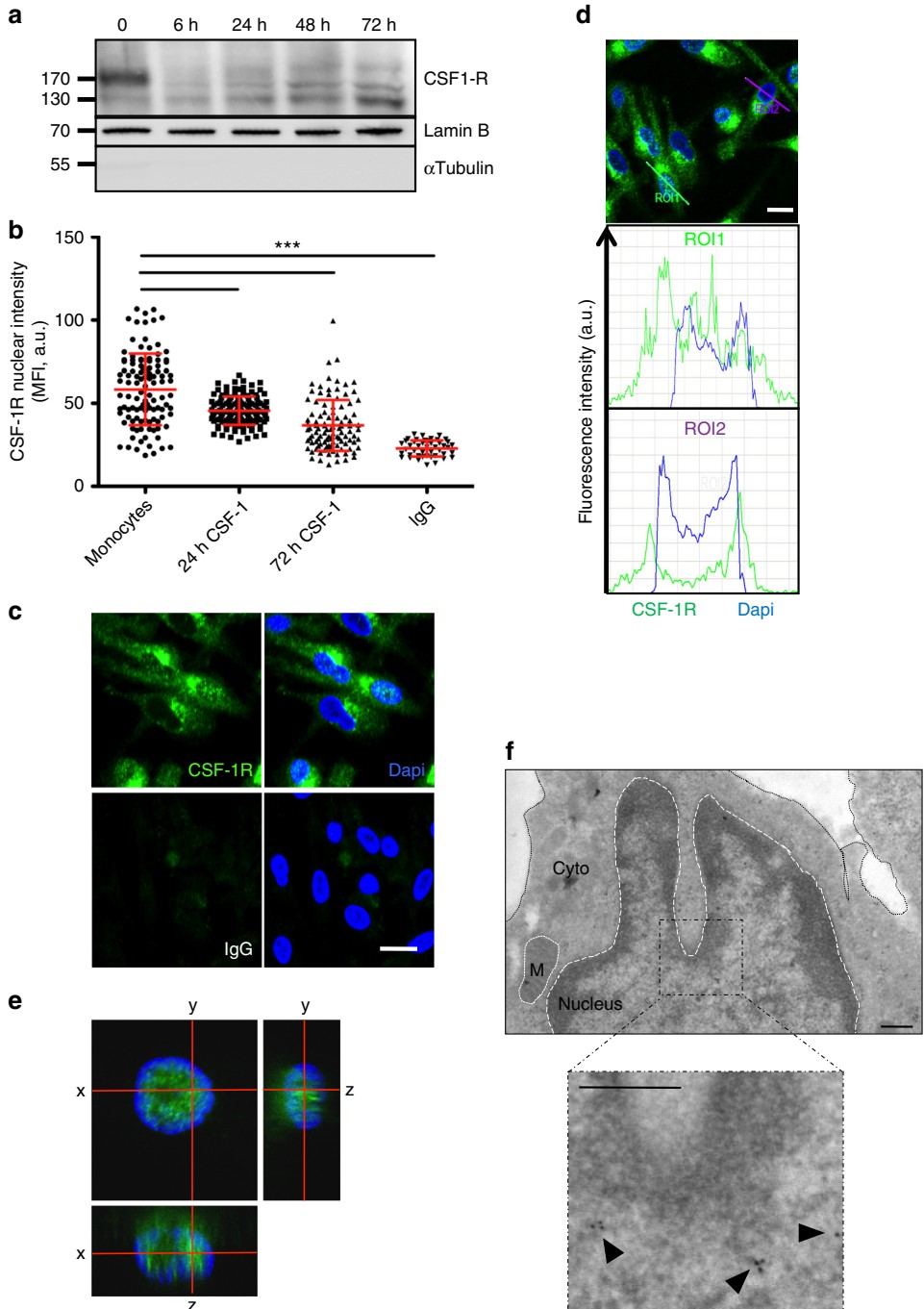

**Fig. 6** Differential CSF-1R chromatin localization in CSF-1-induced macrophages. **a** Nuclear proteins were extracted from monocytes stimulated with 100 ng/mL CSF-1 during the indicated times, and analyzed by immunoblotting for CSF-1R, Lamin B (nucleus marker) and α-tubulin (cytosol marker) ($n = 3$). **b** Monocytes were stimulated or not with 100 ng/mL CSF-1 during the indicated times, stained with an anti-CSF-1R antibody (Cter sc-692) or a control IgG and Dapi, followed by confocal imaging analysis of CSF-1R nuclear localization (MFI, mean fluorescence intensity, A.U. arbitrary units, mean $+/-$ SD, Paired $t$ test: ***$P < 0.001$, $n = 1$). **c** Monocytes were treated 3 days with 100 ng/mL CSF-1, fixed and stained for CSF-1R or control IgG (green) and Dapi (blue) followed by confocal imaging ($n = 3$, scale: 10 μm). **d** Quantification of the signal generated by CSF-1R labeling (green) and Dapi labeling (blue) according to indicated axes (A.U.: arbitrary units, ROI: region of interest, scale: 10 μm). **e** Macrophages were stained with an anti-CSF-1R antibody (C-ter sc-692) and Dapi (blue), followed by confocal imaging analysis (stack of 50 pictures of 0.2 μm) to reconstitute an orthogonal view (scale: 10 μm). **f** Monocytes were differentiated 3 days with 100 ng/mL CSF-1, fixed and stained for CSF-1R (sc-692) followed by electron microscopy ($n = 1$, scale: 500 nm)

(transcription termination site) and 5′UTR regions compared with unstimulated monocytes (Fig. 8a, b). The fraction of CSF-1R peaks colocalizing with H3K4me3 after 6 h of CSF-1 exposure was 90% (Supplementary Data 13). Only 28% of CSF-1R peaks remained similar to unstimulated cells (Fig. 8c and Supplementary Data 14). CSF-1R was detected on a fraction of chromatin

sites identified in macrophages, e.g., *PU.1* TSS, while some CSF-1R localizations as *CSF2RB* TSS were only transiently detected in cells treated for 6 h with CSF-1 (Fig. 8c). Pathway analysis indicated that CSF-1R localizes mainly to genes involved in CSF-1 signaling, phagocytosis, and migration (Supplementary Table 3). Most of these pathways differ from those identified in resting

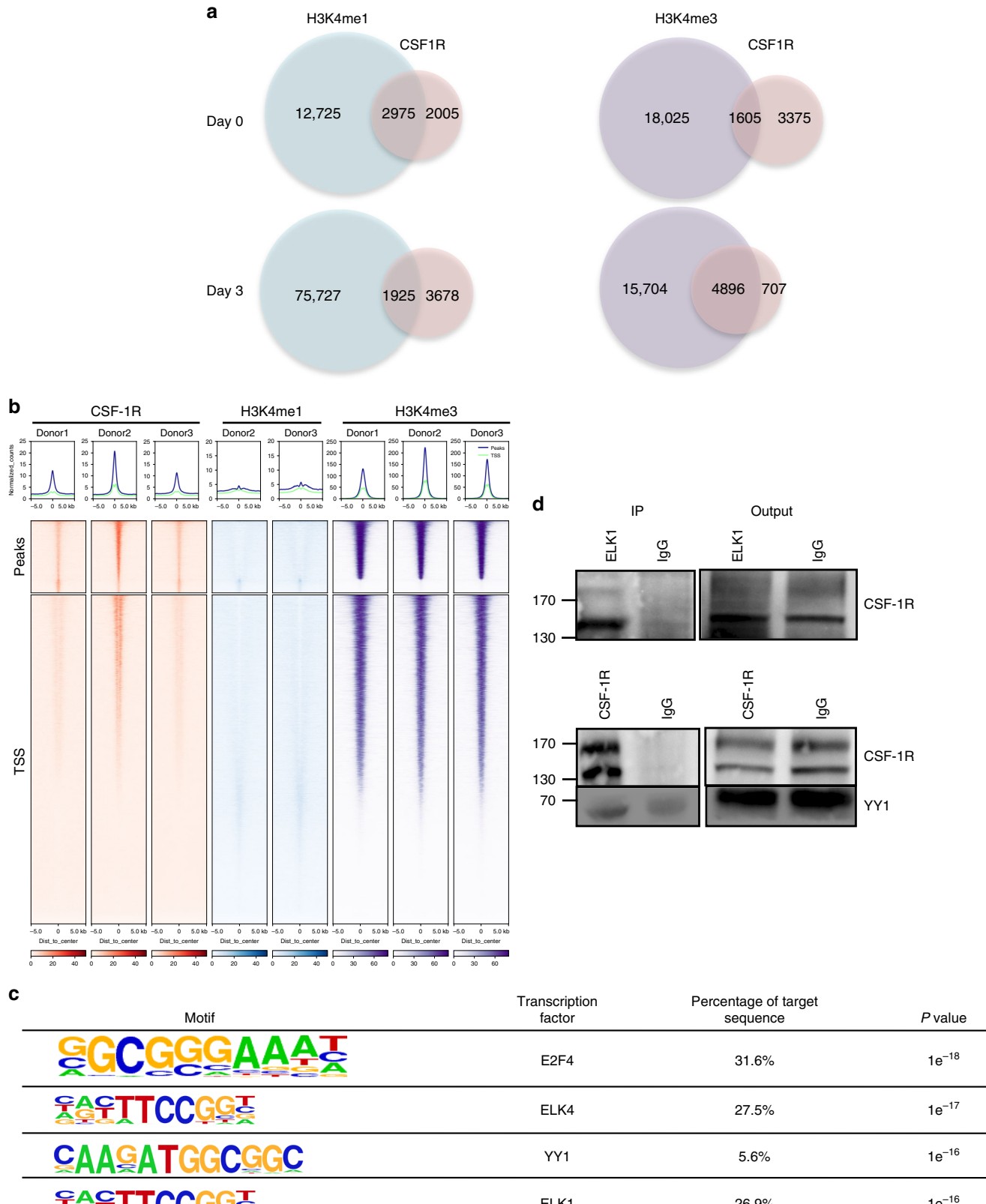

**Fig. 7** Differential CSF-1R chromatin localization in CSF-1-induced macrophages. **a** Venn diagram of ChIP-seq peak number in untreated monocytes (Day 0) and monocytes treated with 100 ng/mL CSF-1 for 3 days (Day 3) showing peaks common to the 3 donors. Left: CSF-1R and H3K4me1 peaks; right: CSF-1R and H3K4me3 peaks. **b** Ranking heatmap centered on CSF-1R peaks (top) and on gene transcription starting sites (TSS) of CSF-1R, H3K4me1 and H3K4me3 peaks in CSF-1-differentiated macrophages (3 days 100 ng/mL). **c** Motif analysis of ChIP-seq experiment on 3 day CSF-1 differentiated monocytes (100 ng/mL). **d** Co-immunoprecipitation of CSF1R and ELK1 (upper panel) or YY1 (lower panel). Immunoprecipitation (IP) was performed with an anti-ELK1 (sc-365876) or anti-CSF-1R (sc-365719) and a control IgG in monocytes treated for 3 days with 100 ng/mL CSF-1, followed by immunoblotting with an anti-CSF-1R ($n = 2$ each) or an anti-YY1 ($n = 3$) antibodies

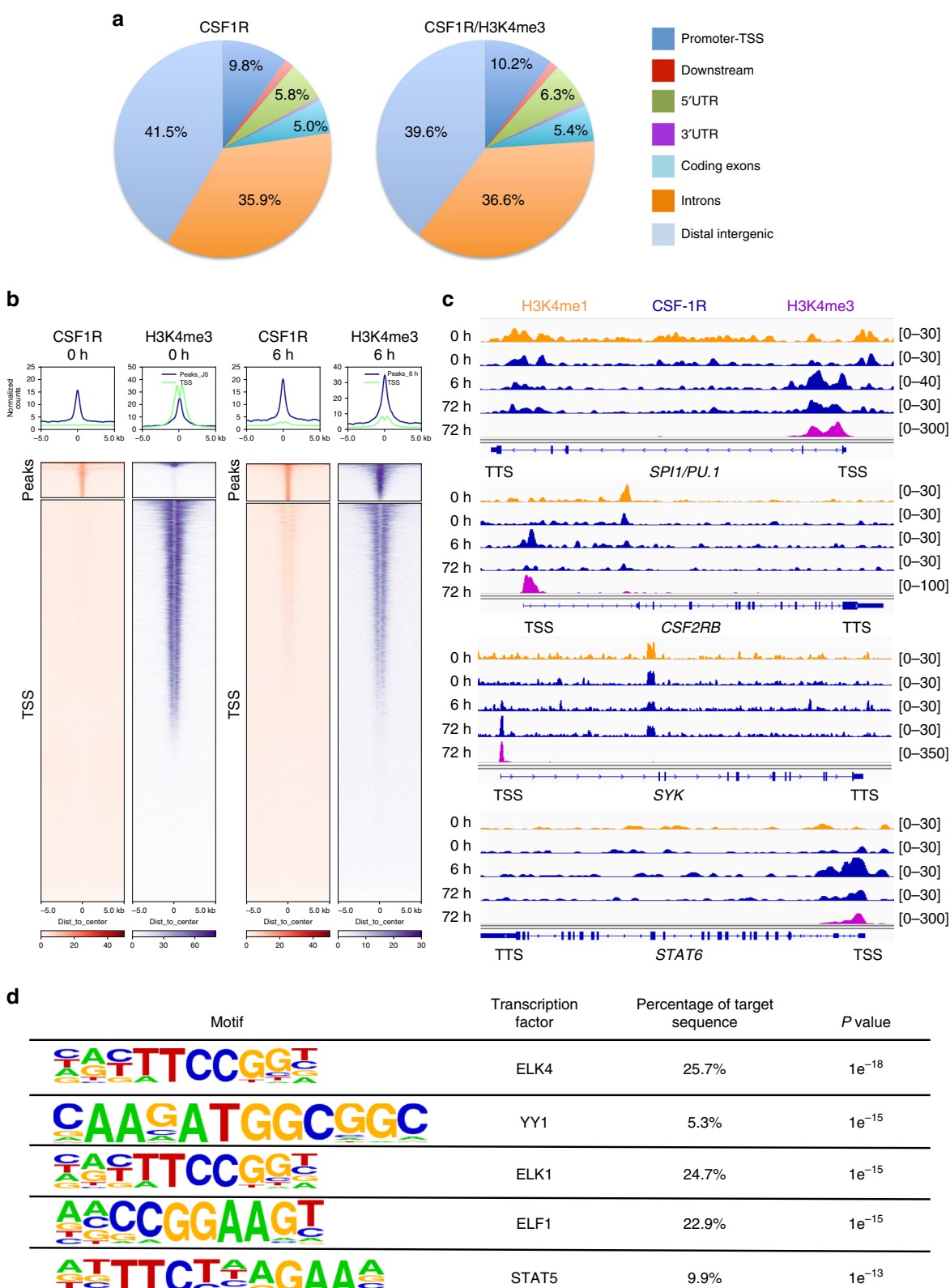

**Fig. 8** CSF-1 modulates CSF-1R recruitment on chromatin within a few hours. **a** Repartition of CSF-1R ChIP-seq peaks and the intersection of CSF-1R and H3K4me3 peaks on the genome of monocytes exposed for 6 h to 100 ng/mL CSF-1. Reference genome is shown on Fig. 3b. **b** Heatmap centered on CSF-1R peaks (top) and on gene transcription starting sites (TSS) of CSF-1R and H3K4me3 peaks in unstimulated monocytes (0 h) and monocytes exposed to CSF-1 for 6 h (6 h). **c** ChIP-Seq experiments showing CSF-1R, H3K4me1 and H3K4me3 peak localization on *PU.1*, *CSF2RB*, *SYK* and STAT6 genes in monocytes, either untreated (0 h) or treated with CSF1 for 6 h (6 h) or 3 days (72 h) (TSS: transcription starting site, TTS: transcription termination site). **d** Motif analysis of the CSF-1R ChIP-seq peaks from monocytes stimulated with 100 ng/mL CSF-1 for 6 h

monocytes, as well as in macrophages. Motif analysis indicated CSF-1R localization on ELK and YY1 motifs, as in macrophages, and ELF1 and STAT5 motifs, which were only transiently detected after 6 h of CSF-1 stimulation (Fig. 8d). These results demonstrate a dynamic CSF-1R/chromatin interaction through monocyte to macrophage differentiation upon CSF-1 exposure.

**Abnormal CSF-1R expression in CMML monocytes.** CMML is a myelodysplatic/myeloproliferative syndrome characterized by an abnormal expansion of circulating classical monocytes[28,29]. We detected a heterogeneous decrease in *CFMS* mRNA and CSF-1R protein expression in patient monocytes compared with healthy donor monocytes (Fig. 9a, b). In many patient samples, the decreased expression of CSF-1R precluded CSF-1R ChIP-seq experiments. In those in which we succeeded in carrying out CSF-1R immunoprecipitation, we generated a bed file of 2276 reproducible peaks (intersection of 4980 peaks obtained with the monoclonal N-terminal CSF-1R antibody with 23,480 peaks obtained with the polyclonal, C-terminal CSF-1R antibody). Some enrichments on chromatin sites were conserved in healthy donors and patient monocytes, e.g., on intergenic area on chromosome 1 and *CEBPA* and *BCL2* genes (Fig. 9c) whereas others were lost in some patient cells, e.g., in *BPIFB1* gene, downstream of *AP1M1* gene and in intergenic areas on chromosome 11 and 22 (Fig. 9d). These results indicated a heterogeneous disruption of CSF-1R interaction with chromatin in CMML patient monocytes.

## Discussion

The present study demonstrates the presence of a fraction of CSF-1R in human primary monocyte nucleus and its recruitment to chromatin where it colocalizes with H3K4me1 marks and interacts with EGR1 to modulate gene expression. Upon CSF-1-induced monocyte differentiation into macrophage, nuclear CSF-1R is relocated on chromatin within a few hours to differentially influence gene expression through interaction with YY1 and ELK transcription factors. Small molecule inhibitors of CSF-1R interfere with these nuclear functions that are otherwise disorganized in CMML dysplastic monocytes.

Several RTKs have been shown to modulate gene transcription by interacting with transcription factors[30,31], carrying these proteins to the nucleus[32], promoting their recruitment to DNA[33], or phosphorylating nuclear proteins[34,35]. CSF-1R itself was shown to be recruited to specific genes in breast cancer cells[19]. We now demonstrate a physiological and dynamic role of CSF-1R in regulating gene expression along the differentiation of human monocytes. These cells, which circulate in the blood stream, are considered as an intermediate developmental stage between bone marrow precursors and some tissue macrophage populations. Nuclear CSF-1R may be a molecular component of monocyte differentiation into macrophages through modulating the expression of key regulatory genes such as *CEBPD* and *KLF6*[36].

CSF-1R staining was diffuse in resting monocytes. The punctual staining detected in response to CSF-1 treatment may indicate internalization in endocytic vesicles following receptor activation at the membrane level, and the formation of protein complexes involving CSF-1R along specific chromatin sites in the nucleus. We noticed a decrease in nuclear CSF1R upon treatment with CSF-1R small molecule inhibitors, suggesting that interaction of the receptor with its ligand may drive its nuclear accumulation. The slight but reproducible increase in nuclear CSF1R observed at highest BLZ945 concentration could indicate a toxic effect of the compound.

A distinct interaction of CSF-1R with chromatin is observed among cell types at the level of individual genes, e.g., CSF-1R is recruited to distinct sequences on *CMYC* and *CJUN* genes in macrophages and breast cancer cells, respectively[19]. As CSF-1R lacks DNA binding domain, it may be recruited to these DNA sequences through interaction with other proteins, including transcription factors. These interactions may change with the functional program followed by these cells, e.g., CSF-1R interaction with EGR1 detected in monocytes is lost in macrophages in which the transcription factor expression is downregulated[36]. Changes in epigenetic landscape along monocyte differentiation may be part of the molecular mechanisms that modulate CSF-1R interaction with nuclear chromatin[22]. An endotoxin challenge would probably modify the pattern of CSF-1R/chromatin interaction in either monocytes or macrophages.

CSF-1R relocalization to active genes with an H3K4me3 mark was detected as soon as 6 h after CSF-1 stimulation without being totally similar to that observed after 72 h of CSF-1 exposure, suggesting a dynamic pattern of CSF-1R recruitment to chromatin. Monocytes are highly plastic cells whose differentiation is diverted towards osteoclasts when RANKL is combined with CSF-1[37], while a combination of CSF-2 and IL-4 promotes myeloid dendritic cell formation[38]. Each of these processes may generate a specific pattern of nuclear CSF-1R interaction with chromatin. Finally, CSF-1R expression level and chromatin recruitment are altered in CMML patient monocytes, which may be related to splicing and epigenetic aberrations that characterize this disease[29,39] and further supports the dysplastic nature of these cells.

Nuclear CSF-1R could also play an important role in CSF-1 capacity to instruct hematopoietic stem cell commitment towards monocyte lineage through promoting *PU.1/SPI1* gene transcription, as depicted in mice[40]. The differential recruitment of CSF1R on *PU.1/SPI1* sequence in monocytes and macrophages suggests a subtle regulation of this gene in response to CSF-1. More generally, CSF-1R nuclear functions could be shared with other RTK subfamily members such as PDGF-R (Platelet-derived growth factor receptor), which was detected in cell nucleus, as well as KIT (Mast/Stem cell growth factor receptor) and FLT3 (Fms-like tyrosine kinase 3), whose nuclear localization remains to be explored. Finally, therapeutic strategies targeting these receptors are currently developed, e.g., CSF-1R-targeting strategies are used to deplete tumor-associated macrophages[12]. Further investigations will indicate how nuclear location of CSF1R could modulate the response to these drugs[10,30].

## Methods

**Antibodies and reagents.** For immunofluorescence, imaging cytometry and electron microscopy experiments, we used anti-CSF-1R antibodies (sc-692, sc-365719), anti ELK-1 antibody (sc-365876) and corresponding isotype controls (normal rabbit IgG sc-2027, normal mouse IgG sc-2025) from Santa Cruz Biotechnology (Clinisciences, Nanterre, France). Secondary antibodies (Alexa Fluor 488 goat anti-Rabbit) were purchased at ThermoFisher Scientific (Waltham, USA) and at TEBU-Bio (Le Perray-en-Yvelines, France) (goat anti-rabbit and anti-mouse with gold particles). For immunoblot experiments, we used Cell Signaling Technology (Clinisciences, Nanterre, France) antibodies: anti-CSF-1R (#3152), anti-YY1 (#2185), anti-HDAC2 (#2540), anti-αTUBULIN (#3873); Santa Cruz Biotechnology antibodies: anti-GAPDH (sc-32233), anti-EGR1 (sc-110), anti-LAMIN B (sc-6217); anti-ACTIN (A5441) from Sigma-Aldrich (Saint-Quentin Fallavier, France) and anti-H3K4me3 from Active Motif (39159) (La Hulpe, Belgium). Secondary antibodies were purchased at ThermoFisher Scientific (anti-rabbit, anti-mouse and anti-goat HRP-conjugated). For immunoprecipitation experiments, we used anti-CSF-1R (sc-365719), anti-ELK1 (sc-365876) or control Mouse IgG (sc-2025) from Santa Cruz Biotechnology. For ChIP-qPCR and ChIP-seq experiments we used two anti-CSF-1R antibodies (sc-692 and sc-46662), EGR1 antibody (sc-110X), corresponding isotype controls (normal rabbit IgG sc-2027, normal mouse IgG sc-2025) from Santa Cruz Biotechnology, anti-H3K4me3 and H3K4me1 from Active Motif (39159 and 39297, respectively). Chemical inhibitors of CSF-1R were purchased at Interchim (BLZ945, ThermoFisher Scientific) and Sigma-Aldrich (GW2580).

**Monocyte sorting and culture.** Peripheral blood mononucleated cells (PBMC) were obtained from healthy donor buffy coats (Etablissement Français du sang, Rungis, France) or newly diagnosed CMML patients whose samples were collected

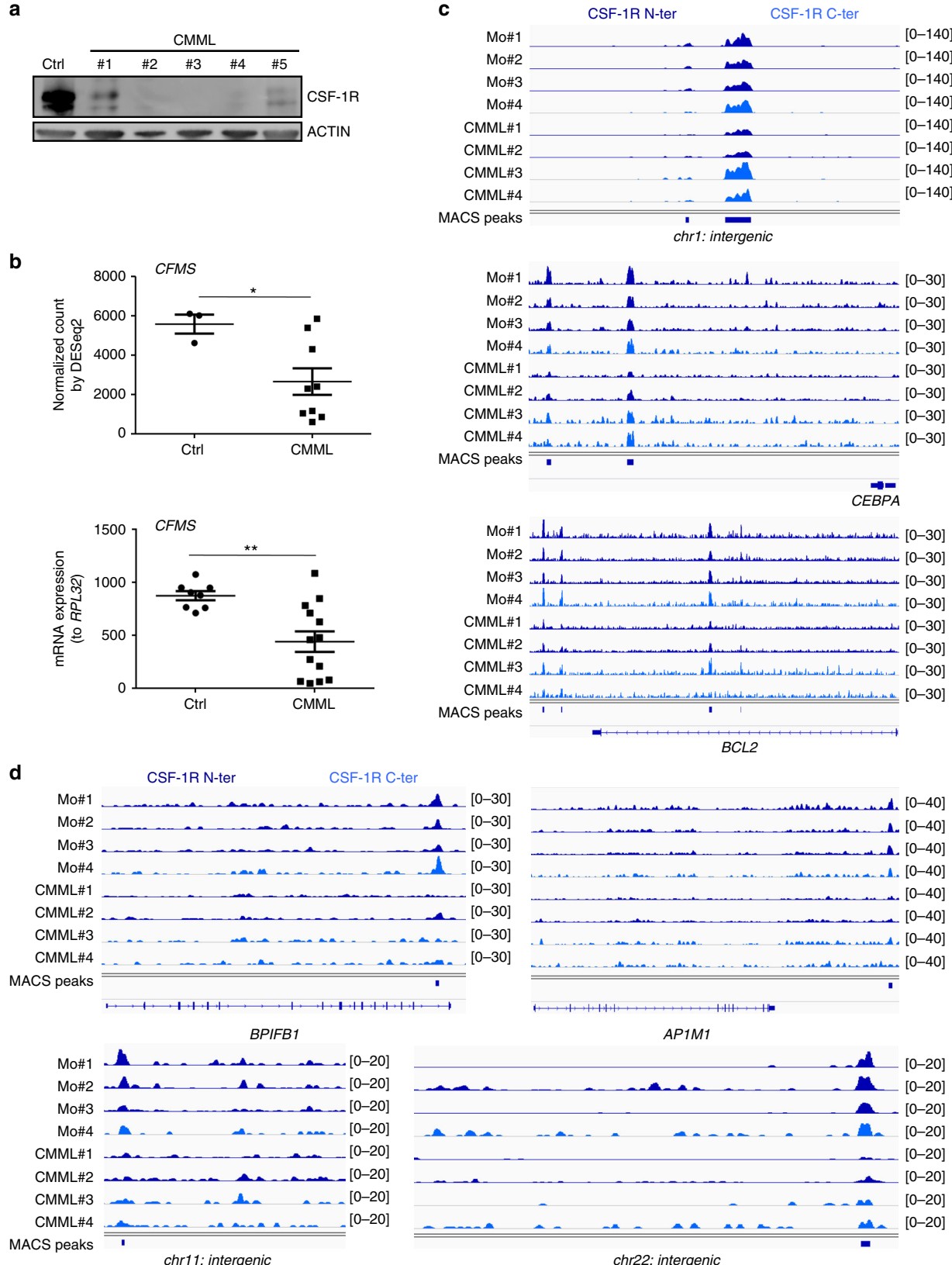

**Fig. 9** CSF-1R is abnormally expressed and recruited on chromatin in CMML monocytes. **a** Total proteins were extracted from isolated control or CMML monocytes and analyzed by immunoblotting with antibodies that recognize CSF-1R and actin. **b** CFMS expression was assessed by RNA-sequencing and qPCR in monocytes of 9 and 13 CMML patients, respectively, compared with 3 and 8 control buffy coats (Ctrl), respectively (mean +/− SEM, Mann–Whitney test: *$P < 0.05$, **$P < 0.01$). **c** ChIP-Seq experiments showing CSF-1R peak localization on intergenic area on chromosome 1 (chr1: intergenic), upstream of *CEBPA* and on *BCL2* genes that are conserved in control and CMML monocytes. **d** ChIP-Seq experiments showing CSF-1R peak localization on *BPIFB1*, downstream of *AP1M1* and on intergenic area on chromosome 11 and 22 (chr11: intergenic and chr22: intergenic) that can be lost in CMML compared with healthy donor monocytes (CSF-1R N-ter: sc-46662 monoclonal antibody in dark blue, CSF-1R C-ter: sc-692 polyclonal antibody in light blue)

with informed consent following the authorization provided by the ethical committee Ile-de-France 1 (DC-2014-2091), separated on Pancoll (Pan-Biotech, Dutscher, Brumath, France). Peripheral blood CD14+ monocytes were sorted with magnetic beads and the AutoMacs system (Miltenyi Biotech, Paris, France)[41]. Monocytes used for nucleofection experiments were sorted with Monocyte isolation kit II (Miltenyi Biotech). After sorting, monocytes were cultured overnight at $10^6$/mL in RPMI 1640 Glutamax medium (ThermoFisher Scientific) supplemented with 10% heat inactivated fetal bovine serum (FBS, Lonza, Amboise, France), 1% penicillin/streptomycin, and 2 mM L-Glutamine (ThermoFisher Scientific). Cells were then seeded at $0.5 \times 10^6$/mL and differentiated into macrophages in the same medium containing recombinant human CSF-1 (100 ng/mL Peprotech, Neuilly-Sur-Seine, France) during 1 to 3 days. Cells were detached using PBS-EDTA and centrifuged at 300×g into dry pellets for further RNA or protein extraction.

**Cell lines.** 293T and THP-1 cell lines were purchased at ATCC (LGC Standards, Molsheim, France). THP-1 were cultured in the same medium used for primary cells and 293T in Dulbecco's Modified Eagle's Medium (ThermoFisher Scientific) supplemented with 10% heat inactivated fetal bovine serum, 1% penicillin/streptomycin, and 2 mM L-Glutamine.

**Confocal microscopy.** $150 \times 10^3$ monocytes were cytospined 5 min at 65×g then fixed 10 min with 4% paraformaldehyde (PFA). For adherent macrophages cultured in chamber slides (Nunc Lab-Tek II, ThermoFisher Scientific), the medium was removed and the cells directly fixed with PFA on the slide. After 3 washes in PBS, cells were incubated with anti-CSF-1R antibody (2 μg/mL) or isotype control (2 μg/mL rabbit IgG) and with blocking peptide if necessary (sc-692P at 2 μg/mL, Santa Cruz Biotechnology) diluted in 0.05% saponin during 1 h at room temperature (RT), followed by wash. The secondary antibody was added (10 μg/mL) with 0.05% saponin during 30 min followed by wash. Slides were mounted using Vectashield mounting medium with Dapi (Vectorlabs, Clinisciences). Specimens were examined on Leica confocal microscopes (TCS SPE and SP8, Leica Microsystems, Nanterre, France). The nuclear CSF-1R fluorescence was quantified using ImageJ software. The nucleus areas were selected in each cell using Dapi staining before measuring the mean intensity of CSF-1R fluorescence inside each nucleus with ROI manager. Live cell imaging movie and orthogonal view were created with ImageJ software.

**CSF-1 chemical modification.** Human recombinant CSF-1 (Peprotech) was coupled with Alexa Fluor 488 using the Alexa fluor 488 Microscale Protein Labeling Kit (ThermoFisher Scientific) according to the supplier instructions.

**Live cell imaging.** Adherent cells cultured in chamber slides were serum-deprived, then stained with 100 ng/mL Hoechst (BD Biosciences, Le Pont de Claix, France) during 30 min, stimulated 15 min with 1 μg/mL fluorescent CSF-1 before wash and living cell observation with a spinning disk microscope (Andor Technology, Belfast, UK) during 10 min (1-min delay between each image).

**Electron microscopy.** Cells were centrifuged and fixed as a pellet with 1.6% glutaraldehyde (Sigma-Aldrich) in 0.1 M phosphate buffer, rinsed and equilibrated in 30% methanol, deposited in a Leica EM AFS2/FSP automatic reagent handling apparatus (Leica Microsystems) and embedded in Lowicryl K4M under UV for 40 h at −20 °C and under UV for 40 h at +20 °C. For immunogold electron microscopy (I-EM), ultra-thin sections were incubated for 1 h at room temperature with the anti-rabbit CSF-1R antibody diluted 1/10 in PBT (1× PBS + 5% BSA + 0,25% triton X-100) and for 30 min with a secondary anti-rabbit antibody coupled to 10-nm gold particles (BBInternational, Cardiff, UK) diluted 1/25 in PBS. Thin-sections were briefly contrasted with uranyl acetate and analyzed with a Tecnai Spirit (FEI, Hillsboro, OR). Digital images were taken with a SIS MegaviewIII charge-coupled device camera (Olympus, Tokyo, Japan).

**Imaging cytometry.** Two to $10 \times 10^5$ cells were fixed with 4% PFA for 10 min at 4 °C, washed with 0.05% saponin and incubated 1 h at 4 °C with primary CSF-1R antibody or control isotype diluted in 0.05% saponin. After washing, cells were incubated with a secondary antibody (goat anti-rabbit AF488, 20 μg/mL) diluted in 0.05% saponin for 1 h at 4 °C, washed, and resuspended in 2% FBS-PBS before adding 200 ng/mL Dapi (BD Biosciences). Samples were analyzed on ImageStream X MK2. Focused cells were selected using Gradient RMS function and analysis was performed on singulet CSF-1R+/Dapi+ cells on IDEAS software.

**Cell fractionation and protein extraction.** For extracting the cytoplasmic/membrane and the nuclear fractions, cells were incubated 15 min on ice in a buffer containing 10 mM HEPES, 10 mM KCl, 0.1 mM EDTA, 0.1 mM EGTA, 1 mM DTT, 0.5 mM PMSF (Buffer 1). NP-40 (0.625%) was added and the cells briefly vortexed. Samples were then centrifuged 1 min at 16,000×g at 4 °C. The supernatant (constiting of cytoplasmic plus membrane fraction) was collected and the pellet rinced with Buffer 1 3 times (centrifugation 1 min at 16,000×g and supernatant discarded). The rinced pellets (consisting of the nuclear fraction) were resuspended in 2× Laemmli sample buffer (Bio-Rad, Marnes-la-Coquette, France)

supplemented with 0.1 M DTT and 1× Protease inhibitor cocktail (Roche) during 15 min with agitation at 4 °C. 4× Laemmli sample buffer (Bio-Rad) supplemented with 0.1 M DTT was added to the cytoplasmic fraction and all the samples were then boiled 10 min at 95 °C. To explore the location of CSF-1R dimers, the same protocol was used without DTT in both buffers. After compartment separation, the samples were sonicated 3 times and 4× Laemmli added to cytoplasmic fraction. Samples were then used without boiling. Extraction of total proteins from CMML peripheral blood sorted monocytes was performed with RNA/DNA/Protein Purification Plus Kit (NorgenBiotek, Interchim, Montluçon, France) according to the supplier instructions. 4× Laemmli, 0.1 M DTT and 1× Protease inhibitor cocktail were added and the samples were boiled 10 min at 95 °C.

**Immunoprecipitation.** Cells were lysed 20 min on ice in a buffer containing 50 mM Tris pH 7.4, 150 mM NaCl, 1% NP-40, 10% Glycerol, 1 mM $Na_3VO_4$ and 1× Protease inhibitor cocktail (100 μL for $10 \times 10^6$ cells). Samples were then centrifuged 15 min at 4 °C at 18,000×g and the supernatant containing the proteins was collected. Ten μg of CSF-1R or ELK1 antibody or negative control IgG were added for $10 \times 10^6$ cells and the samples incubated one night at 4 °C with agitation. One hundred μL of Protein A/G PLUS-Agarose (Santa Cruz biotechnology) were incubated with the samples during 2 h at 4 °C with agitation. The complexes were then precipitated with 1 min centrifugation at 2000×g and the supernatant, consisting of «output» was collected. The remaining beads were washed 5 times with 500 μL of lysis buffer without NP-40. 2× Laemmli with 0.1 M DTT and Protease inhibitor cocktail 1× was added on the beads and outputs. The samples were then boiled 10 min at 95 °C.

**Immunoblotting.** Proteins were separated on polyacrylamide gel and transferred to nitrocellulose membrane (ThermoFisher Scientific). Membranes were blocked with 5% bovine serum albumin in PBS, with 0.1% Tween-20 (Sigma-Aldrich) for 40 min at RT, incubated overnight at 4 °C with the primary antibodies (dilution 1/1000e), washed in PBS-0.1% Tween-20, incubated further with HRP-conjugated secondary antibody (400 ng/mL) for 1 h at RT and washed again before analysis using Immobilon Western Chemiluminescent HPR Substrate system (Millipore, Molsheim, France). The chemiluminescent emission was registered by imageQuant LAS 4000 camera (GE Healthcare Life Science, Vélizy, France). Uncropped scans of key immunoblots are shown in Supplementary Fig. 7.

**Transient transfection and siRNA knockdown.** Small interfering RNA and plasmids were introduced into monocytes by nucleofection (Lonza). Briefly, $5 \times 10^6$ monocytes resuspended in 100 μl nucleofector solution with 1nmol siRNA (CSF1-R (1): 5′-GGCUCAACCUCAAAGUCAUGGUGGA-3′; CSF1-R (2): 5′-GCAUCCGGCUGAAAGUGCAGAAAGU-3′; CSF1-R (3): 5′-GCUCAGCCAGCAGCGUUGAUGUUAA-3′) and 150 ng total plasmids were electroporated. Cells were incubated 24 h with 5 ml of pre-warmed complete medium and 10 ng/mL IL-3 (Peprotech) were subsequently added. 48–72 h after transfection, cells were collected using EDTA for gene expression analysis. We used Stealth siRNAduplex (ThermoFisher Scientific) targeting *CFMS* and Stealth RNAi™ siRNA Negative Control. Small interfering RNAs were introduced into macrophages by transfection with lipofectamin 2000 (ThermoFisher Scientific). Monocytes were cultured for 2 days with CSF-1 (100 ng/mL), medium was then replaced by the same medium without antibiotics, and cells were transfected with 0.1 nmol siRNA for $0.3 \times 10^6$ cells according to the supplier instructions. Fourty-eight hours after transfection, cells were collected using EDTA for gene expression analysis.

**Gene expression analysis.** Total RNA was isolated from primary monocytes or macrophages with RNeasy mini Kit (Qiagen). Reverse transcription was carried out using random hexamers (ThermoFischer Scientific) and reverse transcriptase (Super Script II, ThermoFischer Scientific). Quantitative real-time PCR was performed using SYBR-Green (ThermoFisher Scientific) in an applied biosystem 7500 thermocycler using the standard SYBR-Green detection protocol as outlined by the manufacturers (Applied Biosystems, ThermoFisher Scientific). Briefly, 100–200 ng of total cDNA, 50 nM (each) primers, and 1× SYBR-Green mixture were used in a total volume of 20 μl. Specific primer sequences are provided in Supplementary Table 4.

**ChIP-qPCR and ChIP-seq experiments.** Cells were cross-linked with addition of 1% formaldehyde directly to the culture medium for 10 min at RT with agitation. Fixation was stopped by addition of 125 mM glycin during 5 min at RT with agitation and samples were then washed 2 times in ice-cold PBS before addition of SDS lysis buffer (Millipore, 10 uL per $1 \times 10^6$ cells) supplemented with 1% protease inhibitor cocktail (Active Motif). Samples were vortexed and incubated 15 min on ice before 10 min sonication at 40 W (Covaris S220, Woodingdean, UK). The chromatin immunoprecipitation was carried out using ChIP-it express kit according to manufacturer's instruction (Active Motif). Enriched DNA from ChIP and Input DNA fragments were end-repaired, extended with an 'A' base on the 3′ end, ligated with indexed paired-end adapters (NEXTflex, Bioo Scientific, Proteigene, Saint Marcel, France) using the Bravo Platform (Agilent, Les Ulis, France), size-selected after 4 cycles of PCR with AMPure XP beads (Beckman Coulter, Villepinte, France) and amplified by PCR for 10 cycles more. Fifty-cycle single-end

sequencing was performed using Illumina HiSeq 2000 (Illumina, San Diego, USA) or fifty-cycle paired-end sequencing was done using Illumina NovaSeq 6000.

ChIP-qPCRs were performed in the same way as RT-qPCRs with 2 μL of ChIP or IgG samples instead of cDNA. Specific forward and reverse primers are in Supplementary Table 5.

**ChIP-seq analysis**. Reads were aligned into human genome hg19 with BWA aln (v0.7.5a). Peak calling for H3K4me3 and EGR1 was performed using MACS 2.1 using default options for narrow peaks. For CSF-1R and H3K4me1 ChIP-seq analysis, MACS2.1 was used using the following parameters:–nomodel–broad–extsize 200. Annotation and motif analyses have been done with HOMER (v4.9.1) using a window of 600 bp around peak summit. Bigwig files normalized for sequencing depths have been generated with deeptools v3.1.2. Integrative Genomics Viewer (IGV 2.4.14) was used for representation. Heatmaps of short read coverage across the genomic regions of interest (TSS, peaks) were done using deeptools v3.1.2.

**Plasmids**. For luciferase assays, we cloned EGR1 known motif identified by ChIP-seq analysis for CSF-1R recruitment in a Luciferase reporter system (pGL4.17 vector, Promega, Charbonnières-les-Bains, France). Four EGR1 motif identified sequences (F primer: 5′-CTAGCTGCGTGGGTGTGCG TGGGTGTGCGTGGG TGTGCGTGGGTGA-3′; R: 5′-AGCTTCACCCACGCACACCCACGCACACC CACGCACA CCCACGCAG-3′) were ordered at Eurofins (Les Ulis, France), denatured 5 min at 95 ℃ and hybridized by slow temperature drop before enzymatic cloning. CSF1-R binding PU.1 promoter sequences (PU.1 promoter 1 (complete) F primer: 5′-ATTAGCTAGCTCTACTCATCCCTCCAC-3′; R: 5′-TAA GCTTTCCAGCCGGGCTCCGAG-3′, PU.1 promoter 2 (without CSF-1R recruitment sequence) F: 5′-ATTAGCTAGCGGTAGATGGGTGGATAG-3′; R: 5′-ATTT AGATCTTCCAGCCGGGCTCCGA-3′) were first amplified with PCR from healthy monocyte genomic DNA, purified, sub-cloned in TOPO 2.1 vector (ThermoFisher Scientific) before enzymatic cloning in pGL4.17 vector. Luciferase plasmids were co-transfected with Renilla control reporter vector used for transfection efficiency normalization as described previously[42].

**CRISPR Knockout**. Two EGR1 guide RNA sequences and Cas9 were encoded in V2 CRISPR-GFP or Cherry plasmid (one for each guide). Guide sequences were designed in the first exon of EGR1 with CRISPOR software (http://crispor.tefor.net/) and cloned in one of the two lentiviral vectors: EGR1 guide 1 F: 5′-AAACGG CCGGGTTACATGCGGGGC-3′; R: 5′-CACCGCCCCGCATGTAACCCGGC C-3′, EGR1 guide 2 F: 5′-AAACTCGGCGTAGGCCACTGCTTAC-3′; R: 5′-CACC GTAAGCAGTGGCCTACGCCGA-3′. For lentiviral production, 293 T were co-transfected with plasmid of interest along with pCMV and pMD2.G plasmids using jetPRime reagent (Polyplus transfection, Ozyme) according to the manufacturer instructions. Retroviral particles were collected 48 h after transfection and concentrated by ultracentrifugation. THP-1 cells were co-transduced with a MOI of 2.5 with both guides and single-cell sorted based on their positive GFP and Cherry expression (BD Influx). EGR1 knockout was assessed by PCR and Sanger sequencing with the following primers: F: 5′-ATAGAGGCGGATCCGGGGAGT C-3′; R: 5′-GAAACCCGGCTCTCATTCTAAGATC-3′.

**Statistical analysis**. All statistical analyses (Paired and unpaired $t$ tests and Mann–Whitney test for Ctrl versus CMML *CFMS* expression) were performed using prism software.

**Gene ontology (GO) analysis**. Gene ontology annotation was performed using geneontology.org website. We selected the most accurate pathways based on higher fold change.

**Patients**. Peripheral blood samples were collected on EDTA from 28 patients with a CMML diagnosis according to the World Health Organization 2016 criteria[43]. All the procedures were approved by the institutional board of Gustave Roussy and the ethical committee Ile de France 1 (DC-2014-2091, and written informed consent was obtained from each patient. Cohort characterization is summarized in Supplementary Table 6.

**Reporting summary**. Further information on research design is available in the Nature Research Reporting Summary linked to this article.

## Data availability

Datasets are available in the ArrayExpress database at EMBL-EBI (www.ebi.ac.uk/arrayexpress). CSF-1R, H3K4me1, H3K4me3 ChIP-Seq comparing healthy donor monocyte and macrophage profiles (BC212: donor1; BC214: donor2; BC215: donor3), EGR1 ChIP-seq in healthy monocytes (BC275 and BC276) and CSF-1R ChIP-seq using Cterminal antibody in monocyte from healthy (BC270) and CMML (CMML1941: CMML#1982: CMML#2) monocytes are referenced as E-MTAB-6305 and CSF-1R ChIP-seq in THP1 WT clones, THP1 CRISPR clones and in CMML_2130 (CMML#3) and CMML_2609 (CMML#4) using N terminal antibody are referenced as E-MTAB-7756. Bed files of CSF-1R peaks shared by the three donors in monocytes (d0) and

macrophages (d3) are provided in Supplementary Data 15 and Supplementary Data 16. All other relevant data supporting the key findings of this study are available within the article and its Supplementary Information files or from the corresponding authors upon reasonable request. A reporting summary for this Article is available as a Supplementary Information file.

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

## Acknowledgements

This work was supported by INSERM, the Fondation ARC and the Ligue contre le cancer (labeled team). We are grateful to F. Ling and M. Gaudry who provided us sequencing primers for Luciferase cloning assessment, to M. Gastou for flow imaging and cellular compartment separation protocol, to F. Porteu, J.L Villeval, and F. Louache for scientific discussions. We thank Gustave Roussy institute facilities for imaging (Sophie Salome-Desnoulez) and genomics, and the Imagine institute facilities for flow imaging.

## Author contributions

L.B. designed, performed, analyzed the experiments, and drafted the manuscript, C.L. and K.D.M. performed bioinformatic analyses, J.R. provided assistance for cloning experiments, F.D.L. provided assistance for live cell imaging, M.D. for flow imaging, G.P. and S.S. for electron microscopy, A.I. and A.N. for ChIP-seq and RNA-seq, M.M. for managing patient samples, N.D. for qPCR experiments. E.S. and N.D. co-supervised the project and wrote the manuscript.

## Additional information

**Competing interests:** The authors declare no competing interests.

