## [Peer Review File · Nature Communications]

Reviewers' comments:

Reviewer #1 (Remarks to the Author):

The authors present evidence that the receptor for Colony Stimulating Factor 1 (CSF-1R) serves as a transcriptional co-regulator in human monocytes and macrophages. After confirming nuclear presence of CSF-1R by several methods (Figures 1, 2 and 5), they carried out chromatin immunoprecipitation and sequencing (ChIP-Seq) in monocytes and macrophages. In monocytes CSF-1R co-localized with H3K4me1 marks, and sequence motifs for EGR, HOX and GATA proteins enriched under CSF-1R peaks (Figure 3). By contrast, in macrophages CSF-1R co-localized with H3K4me3 marks and sequence motifs for YY1 and ELK proteins (Figure 6) were identified within CSF-1R peaks. Down-regulation of CSF-1R in monocytes activated gene transcription, whereas in macrophages it reduced gene expression (Figures 4, 6). ChIP-Seq of CSF-1R in CMML patient monocytes showed a different pattern of genomic binding compared to monocytes from healthy donors (Figure 8). The authors conclude that CSF-1R acts as a co-repressor in monocytes and a co-activator in macrophages, and is recruited to the genome by different transcription factors in each cell type.

These observations are intriguing and the implications for monocyte biology profound. The critical issue is whether the ChIP-Seq studies with anti-CSF-1R antibodies are robust. I am "on the fence" with regard to this question. The ChIP-Seq profiles shown are quite weak and I presume these are the best of the lot. It is also of concern that only 3% (or 2% in macrophages) of CSF-1R peaks are common to the 3 donors, even though I understand that primary human samples can vary considerably. And, it is unclear whether the motif searches utilized these shared peaks or all peaks. Furthermore, the co-association studies between CSF-1R and various transcription factors is not terribly convincing, and the transcriptional effects of CSF-1R knock-down are not strong. Finally, these studies provide little by way of mechanism for the transformation of CSF-1R from a co-repressor to a co-activator during macrophage differentiation. Because the observations could be quite important, I suggest that the authors firm-up their ChIP-Seq studies to convincingly demonstrate CSF-1R recruitment to DNA. One path could be to use their peptide inhibitor to eliminate non-specific effects during ChIP. Thereafter, they could focus on the most robust and reproducible peaks to query biological function.

Reviewer #2 (Remarks to the Author):

The authors' goal was to determine a role for nuclear CSF-1R in controlling the transcriptional response to CSF-1 and its differential effects throughout monocyte-macrophage maturation. This work may be relevant to a large audience as targeting mechanisms that control monocyte differentiation and macrophage survival through CSF1R is currently of broad research and clinical interest. Overall this paper seems well structured, but I have a few points of concern

-To confirm through imaging that CSF1R can be found in the nucleus under resting conditions, it would help if the authors also showed orthogonal views of the panels in Figure 1A and 1B or did some 3D projection.

-The authors discuss the presence of transient CSF1R dimers in the membrane and cytoplasm with CSF-1 treatment, but not in the nucleus. They also discuss the translocation of CSF1R into the nucleus as being driven by CSF-1 as it is found to colocalize with CSF-1 after CSF-1 treatment. However, they do not seem to discuss the difference in diffuse staining without CSF-1 treatment (as in Figure 1) and punctal staining of CSF-1R in both the membrane and nucleus with CSF-1 treatment (as in Figure 2). A discussion of this point in relation to CSF-1R's functions at the membrane or in the nucleus would help clarify the importance of this obvious visual difference.

-It appears that at the highest concentration of BLZ945 the nuclear CSF-1R starts to actually increase over the previous concentrations that lowered this value. Some discussion of this point would help.

-There should be a graph of nuclear CSF-1R for Figure 5C as there was in Figure 2. Preferably showing this over the time of CSF-1 treatment and not just at the 72 hour time points would help to support the IP results in 5A.

-If the authors could compare the same genes in their siRNA experiments between monocytes and macrophages as in Figure 4E and supplementary Figure 5E, this would help support their point that CSF-1R acts differently on genes depending on the cell's differentiation status.

Reviewer #1

Comments from the reviewer: The authors present evidence that the receptor for Colony Stimulating Factor 1 (CSF-1R) serves as a transcriptional co-regulator in human monocytes and macrophages. After confirming nuclear presence of CSF-1R by several methods (Figures 1, 2 and 5), they carried out chromatin immunoprecipitation and sequencing (ChIP-Seq) in monocytes and macrophages. In monocytes CSF-1R co-localized with H3K4me1 marks, and sequence motifs for EGR, HOX and GATA proteins enriched under CSF-1R peaks (Figure 3). By contrast, in macrophages CSF-1R co-localized with H3K4me3 marks and sequence motifs for YY1 and ELK proteins (Figure 6) were identified within CSF-1R peaks. Down-regulation of CSF-1R in monocytes activated gene transcription, whereas in macrophages it reduced gene expression (Figures 4, 6). ChIP-Seq of CSF-1R in CMML patient monocytes showed a different pattern of genomic binding compared to monocytes from healthy donors (Figure 8). The authors conclude that CSF-1R acts as a co-repressor in monocytes and a co-activator in macrophages, and is recruited to the genome by different transcription factors in each cell type.

These observations are intriguing and the implications for monocyte biology profound. The critical issue is whether the ChIP-Seq studies with anti-CSF-1R antibodies are robust. I am "on the fence" with regard to this question. The ChIP-Seq profiles shown are quite weak and I presume these are the best of the lot. It is also of concern that only 3% (or 2% in macrophages) of CSF-1R peaks are common to the 3 donors, even though I understand that primary human samples can vary considerably. And, it is unclear whether the motif searches utilized these shared peaks or all peaks. Furthermore, the co-association studies between CSF-1R and various transcription factors is not terribly convincing, and the transcriptional effects of CSF-1R knock-down are not strong. Finally, these studies provide little by way of mechanism for the transformation of CSF-1R from a co-repressor to a co-activator during macrophage differentiation. Because the observations could be quite important, I suggest that the authors firm-up their ChIP-Seq studies to convincingly demonstrate CSF-1R recruitment to DNA. One path could be to use their peptide inhibitor to eliminate non-specific effects during ChIP. Thereafter, they could focus on the most robust and reproducible peaks to query biological function.

Issue # 1: The ChIP-Seq profiles shown are quite weak and I presume these are the best of the lot.

Our answer: We agree with the referee that ChIP-seq profiles can be considered as relatively weak, which may be related to the low amount of CSF-1R present in the nucleus (See Fig. 1d, immunoblotting experiments). Confidence in these results came with the reproducible detection of strong peaks. We had chosen to focus our detailed analyses on those located on genes involved in myeloid differentiation and monocyte/macrophage functions such as *PU.1*. We agree that these specific peaks were not the strongest but they were reproducible and associated with monocyte functions and differentiation. We provide below examples of reproducibly detected stronger peaks (three independent experiments):

On these figures, we show CSF-1R peaks (in blue) and inputs (in black) detected on *TCEA3* gene and intergenic region in monocytes from three healthy donors analyzed independently.

On this other figure, we show CSF-1R peaks (in blue) and inputs (in black) detected on *S100A10* gene in monocytes from three healthy donor monocytes analyzed independently after 3 days in culture with 100ng/mL CSF-1.

Issue # 2: It is also of concern that only 3% (or 2% in macrophages) of CSF-1R peaks are common to the 3 donors, even though I understand that primary human samples can vary considerably. And, it is unclear whether the motif searches utilized these shared peaks or all peaks.

Our answer: The low fraction of common peaks in monocytes and in macrophages is probably less related to the use of primary cells with inter-individual variations than to the previous point raised by the referee, *i.e.* the relative weakness of ChIP-Seq profiles that provided a lot of background noise. The following part of our work was focused on common shared peaks, not on all peaks. One of the modifications we introduced in the revised manuscript is the indication that we focused our motif analyses on these common, shared peaks.

In the manuscript we added (in blue): « Motif analysis of ChIP-seq data, performed by focusing on peaks shared by the three donors, indicated that CSF-1R could be recruited on several transcription factor binding sites, including EGR1, EGR2, ARNT/HIF-1 β , FOXA1, HOXA2 and GATA3 motifs (Fig. 3e). »

We also performed ChIP-seq on two additional healthy donors. The number of reads is too low to include these donors in a global analysis but we focused our analysis on common peaks identified in the first series of experiments, which were all detected. For example, we detected similar peaks as in the three first samples on EGR1, EGR2, ARNT and GATA3 sequences. Examples are shown below, showing CSF-1R peaks (in blue) on *SPI1/PU.1* and *KLF6* genes in monocytes collected from five donors (1,2,3 are those included in the initial manuscript, 4 and 5 are two additional healthy donor samples with lower coverage whose results are provided to the reviewer).

We also performed additional ChIP-seq experiments with these two additional healthy donor monocyte samples treated with CSF1 for 3 days to induce monocyte differentiation into macrophages. Again, we detected peaks that were common to the five samples, as shown in the example below provided to the referee:

Issue # 3: Furthermore, the co-association studies between CSF-1R and various transcription factors is not terribly convincing

Our answer: We performed new co-immunoprecipitation experiments of CSF1R and YY1 that replaced the previous panel in figure 6e. We have also added to the manuscript a

new experiment using immunofluorescence and confocal microscopy to further support an interaction between CSF-1R and ELK1 in CSF-1-induced macrophages. We included this experiment in the manuscript as supplementary figure 7e.

The text was modified as follows: « ELK proteins bind to serum response elements to modulate immune response^{24,25}, while YY1 is involved in macrophage response to lipopolysaccharides²⁶. Co-immunoprecipitation (Fig. 6 d,e) and immunofluorescence (Supplementary Fig. 7e) experiments in macrophages argued for CSF-1R interaction with these 2 proteins.»

Monocytes were treated 3 days with 100ng/mL CSF-1, fixed and stained for CSF-1R (green), ELK1 (red) or control IgG and Dapi (blue) followed by confocal imaging (n = 2, scale : 10µm)

Issue # 4: I suggest that the authors firm-up their ChIP-Seq studies to convincingly demonstrate CSF-1R recruitment to DNA. One path could be to use their peptide inhibitor to eliminate non-specific effects during ChIP.

Our answer: This comment completes issues #1 and #3 and we thank the referee for the suggestion. To confirm CSF-1R ChIP-seq results, we had validated CSF-1R peak localization on chromatin by ChIP-qPCR in 3 to 4 additional donors (as shown in supplemental figure 3a and 3b) and with two different antibodies: the first antibody, directed against the N-terminal region of CSF-1R, was used for ChIP-seq experiments; the second antibody, which recognizes the C-terminal part of CSF-1R, validated the results obtained with the first one. Of note, we obtained less DNA after chromatin immunoprecipitation in CSF-1-differentiated macrophages than in primary monocytes, which correlates with the decreased level of CSF1R in macrophage nucleus (as shown in figure 5a). The relocation of CSF1R within a few hours following monocyte stimulation with CSF-1 further argues for the specificity of our ChIP-seq studies.

To further enforce these results, we treated monocytes during 6h with 50nM BLZ945 (or control DMSO), which decreases CSF-1R nuclear localization. Then, we performed ChIP-qPCR to evaluate if this inhibitory molecule could decrease CSF-1R recruitment on chromatin, which was indeed observed on 5 studied genes, as summarized below.

There was no blocking peptide available to antagonize the CSF1R antibody used for ChIP-seq experiments and the use of CSF-1R siRNA would have completely impaired the ChIP-seq library preparation. Therefore, we have chosen another approach, i.e. to decrease EGR1 expression and determine if it could decrease CSF-1R recruitment on chromatin. This approach could also provide an additional answer to the issue # 3 raised by the referee regarding the CSF1R/EGR1 interaction.

Down regulation of EGR1 in monocytes using siRNA was not efficient enough. Therefore, we moved to a CRISPR-Cas9 approach in the THP-1 monocytic cell line. Preliminary experiments had shown that CSF-1R peaks that co-localize with EGR1 in monocytes were also detected in THP-1 cells. We obtained three clones with homozygous deletion of EGR1 and performed CSF-1R ChIP-seq experiments in these clones as well as their wild-type counterpart. Importantly, the quantity of DNA captured by the anti-CSF1R antibody in EGR1-deleted clones was very low as compared to wild type cells. Consequently, the number of reads obtained by sequencing individual EGR1-deleted clones precluded any bioinformatics analysis. We had to pool the reads from three EGR1-deleted clones to observe the loss of CSF-1R peaks on EGR1 motifs. Altogether, a loss of EGR1 in THP-1 prevents CSF-1R recruitment on EGR1 motifs, further arguing for a role for this transcription factor in CSF-1R recruitment on monocyte DNA.

We added these results to the revised version of the manuscript by adding a new paragraph and new figures (Fig. 4e and supplementary Fig. 5): «To further explore the role of EGR1 in recruiting CSF-1R at the chromatin level, we deleted EGR1 in THP1 monocytic cell line using Crisper/Cas9 technology. Preliminary experiments had shown that 2,421 CSF-1R peaks common to primary human monocytes and THP1 cell line, including peaks that co-localize with EGR1. We obtained three clones with *EGR1* homozygous deletion, which was validated by Sanger sequencing (Supplementary Fig. 5a) and RT-qPCR (Supplementary Fig. 5b), before performing CSF1R ChIP-seq analysis in these clones and their wildtype counterpart. Importantly, the quantity of DNA captured by the anti-CSF1R antibody in EGR1-deleted clones was very low as compared to wild type cells (Supplementary Fig. 5c). Nevertheless, we deep sequenced the totality of these libraries and pooled them for analysis, showing that EGR1 deletion abrogates CSF-1R localization at EGR1 sites on the chromatin, for example on *PU.1*, *CALML5*, *CEBPD*, *TLR10* and *ROR2* genes (Fig. 4e and Supplementary Fig. 5d).»

To report these new results, we added information in the material and methods section :

Cell lines

293T and THP-1 cell lines were purchased at ATCC (LGC Standards, Molsheim, France). THP-1 cells were cultured in the medium used for primary cells and 293T in Dulbecco's Modified Eagle's Medium (ThermoFisher Scientific) supplemented with 10% heat inactivated fetal bovine serum, 1% penicillin/streptomycin and 2mM L-Glutamine.

CRISPR Knockout

Two EGR1 guide RNA sequences and Cas9 were encoded in V2 CRISPR-GFP or Cherry plasmid (one for each guide). Guide sequences were designed in the first exon of *EGR1* with CRISPOR software and cloned in one of the two lentiviral vectors:

EGR1 guide 1

F: 5'-AAACGGCCGGGTTACATGCGGGGC-3'

R: 5'-CACCGCCCCGCATGTAACCCGGCC-3'

EGR1 guide 2

F: 5'-AAACTCGGCGTAGGCCACTGCTTAC-3'

R: 5'-CACCGTAAGCAGTGGCCTACGCCGA-3'

For lentiviral production, 293T were co-transfected with the plasmid of interest along with pCMV and MD2G plasmids using jetPRime reagent (Polyplus transfection, Ozyme) according to manufacturer's instructions. Retroviral particles were collected 48h after transfection and concentrated by ultracentrifugation. THP-1 cells were co-transduced with a MOI of 2.5 with both guides and single-cell sorted based on their positive GFP and Cherry expression (BD Influx). *EGR1* knockout was assessed by PCR and Sanger sequencing with the following primers:

F: 5'-ATAGAGGCGGATCCGGGGAGTC-3';

R: 5'-GAAACCCGGCTCTCATTCTAAGATC-3'.

New Supplemental figure 5: *EGR1* is involved in CSF-1R recruitment on chromatin in THP-1 cells. **a.** Sequencing of *EGR1* exon 1 in THP-1 clones (#1, #2 and #3) deleted CRISPR-Cas9 mediated deletion. Sequences are aligned against wild-type *EGR1* sequence (bottom line, start codon in yellow). **b.** mRNA expression (qPCR) of *EGR1* gene in wild-type and *EGR1*-deleted clones (mean +/- SD of 3). **c.** Migration profile (Agilent Bioanalyzer) of Illumina libraries built using DNA immunoprecipitated with anti-CSF-1R antibody (sc-46662) in wild-type (WT) and *EGR1*-deleted clones. **d.** Peak calling on *CEBPD*, *TLR10* and *ROR2* genes for *EGR1* in monocytes and for CSF-1R in two healthy donor monocyte samples (Mo), two wild-type (WT) THP-1 clones (in pink), and the pool of 3 *EGR1*-deleted THP-1 clones (in orange).

Reviewer #2

Comments from the reviewer: The authors' goal was to determine a role for nuclear CSF-1R in controlling the transcriptional response to CSF-1 and its differential effects throughout monocyte-macrophage maturation. This work may be relevant to a large audience as targeting mechanisms that control monocyte differentiation and macrophage survival through CSF1R is currently of broad research and clinical interest. Overall this paper seems well structured, but I have a few points of concern

Issue # 1: To confirm through imaging that CSF1R can be found in the nucleus under resting conditions, it would help if the authors also showed orthogonal views of the panels in Figure 1A and 1B or did some 3D projection.

Our answer: We thank the referee for her/his suggestion that enforces the demonstration of CSF1R localization in the nucleus of resting monocytes and macrophages, and now provide orthogonal views of monocytes and macrophages as new panels in Fig. 1c and Fig. 5e, respectively.

We added the related information in the revised manuscript:

«We sorted human monocytes from healthy donor peripheral blood and detected CSF-1R in both the cytoplasm and the nucleus by confocal microscopy (Fig. 1a,b), which was further confirmed by orthogonal views (Fig. 1c).»

«Confocal imaging also detected CSF-1R in the nucleus of a fraction of these macrophages (Fig. 5c,d), which was confirmed by orthogonal views (Fig. 5e) and further validated by immunogold staining and electron microscopy (Fig. 5f). »

Issue # 2: However, they do not seem to discuss the difference in diffuse staining without CSF-1 treatment (as in Figure 1) and punctal staining of CSF-1R in both the membrane and nucleus with CSF-1 treatment (as in Figure 2). A discussion of this point in relation to CSF-1R's functions at the membrane or in the nucleus would help clarify the importance of this obvious visual difference.

Our answer:

This is a very good point and, indeed, we had not discussed it. We propose the following changes in the revised manuscript, underlying the punctual staining in the results to discuss that point later in the manuscript. In the cytoplasm, this may be due to the concentration of the activated and internalized receptor in endocytic vesicles. In the nucleus, we also observed such aggregation of CSF1R in electron microscopy. Although we have no definitive explanation at this step, this could be due to the concentration of the receptor at specific chromatin sites, forming complexes with transcription factors.

Changes are shown here in blue :

« Peripheral blood monocytes were exposed to AF488-labeled recombinant CSF-1 for 15min before fixation and staining with anti-CSF-1R antibody. As expected, CSF-1 and CSF-1R co-localized mainly at the plasma membrane and in the cytoplasm. CSF-1 was also detected in monocyte nucleus where it co-localizes with CSF-1R (Fig. 2a and movie as supplementary Fig. 2). **Of note, CSF-1R staining after CSF-1 treatment was more punctual compared to resting monocytes (Fig. 1a).** This nuclear localization of CSF-1 and CSF-1R could not be related to nuclear localization signals (NLS) as CSF-1R primary sequence is devoid of this sequence and the putative NLS (amino acids 521 to 524) in CSF-1 sequence¹⁹ is deleted from the recombinant CSF-1 used in this experiment. CSF-1 nuclear accumulation could be prevented by monocyte pre-treatment for 3h with small molecule CSF-1R inhibitors, either BLZ945 or GW2580 (Fig. 2b). Confocal imaging further showed that monocyte exposure to CSF-1R inhibitors for 3 hours partially depleted nuclear CSF-1R **in a dose-dependent manner** (Fig. 2c), which could be prevented by leptomycin B, an inhibitor of CRM1-mediated nuclear

export (Fig. 2d). All together, these results suggest a role for CSF-1 in CSF-1R nuclear localization in human monocytes. »

In the discussion : « While CSF-1R staining was diffuse in resting monocytes, a punctual staining was detected in monocytes exposed to CSF-1R concentration, which may indicate internalization and concentration in endocytic vesicles following receptor activation at the membrane level, and the formation of protein complexes involving CSF-1R along specific chromatin sites in the nucleus.»

Issue # 3: It appears that at the highest concentration of BLZ945 the nuclear CSF-1R starts to actually increase over the previous concentrations that lowered this value. Some discussion of this point would help.

Our answer: We agree with the reviewer's comment. Although it appears to be less pronounced with other samples, we still observed a slight increase of CSF-1R intensity at 1 μ M BLZ945 when we pooled the results of 3 independent experiments (see below). We do not have a clear explanation to that effect at this step. We have changed the text in the revised version of the results by removing the "dose-dependent effect" : « Confocal imaging further showed that monocyte exposure to CSF-1R inhibitors for 3 hours partially depleted nuclear CSF-1R ~~in a dose-dependent manner~~ (Fig. 2c), which could be prevented by leptomycin B, an inhibitor of CRM1-mediated nuclear export (Fig. 2d). »

Monocytes from 3 donors were treated or not with indicated concentrations of BLZ945 or GW2580 (DMSO as negative control) before confocal imaging analysis of CSF-1R nuclear localization (MFI, mean fluorescence intensity, A.U., arbitrary unit, mean +/- SD, ***P<0.001, n = 3)

And to add in the discussion : « We noticed a slight but significant increase in nuclear CSF-1R when monocytes were treated with CSF-1R small molecule inhibitor BLZ945 at 1 μ M compared to lower concentrations that decreased nuclear CSF-1R. This effect, whose origin remains unclear, suggests that an optimal concentration of inhibitory molecule will be requested to optimally reduce nuclear CSF-1R»

Issue # 4: There should be a graph of nuclear CSF-1R for Figure 5C as there was in Figure 2. Preferably showing this over the time of CSF-1 treatment and not just at the 72 hour time points would help to support the IP results in 5A.

Our answer: As suggested, we quantified CSF-1R nuclear staining over the time after CSF-1 treatment by two different approaches (immunofluorescence and flow imaging)

and the results observed were in accordance with immunoblot experiments. Quantification of immunofluorescence was performed on results summarized in Fig. 1a,b and Fig.5 b,c

We propose to add CSF-1R nuclear staining over the time after CSF-1 treatment in the manuscript as Fig. 5b and to modify the text as follows: « CSF-1R nuclear staining was quantified over the time after CSF-1 treatment by confocal imaging, showing again a decrease of CSF-1R nuclear localization after 24h and 72h compared to resting monocytes (Fig. 5b). »

Monocytes from the donor whose results are shown on Fig. 1 and 5 were stimulated or not with 100ng/mL CSF-1 for indicated times, stained with an anti-CSF-1R antibody (Cter sc-692) or a control IgG and Dapi, followed by confocal imaging analysis of CSF-1R nuclear localization (MFI, mean fluorescence intensity, mean +/- SD, ***P<0.001, n = 1).

In addition we performed Amnis analyses with two different antibodies

Unstimulated monocytes or monocytes exposed to 100ng/mL CSF-1 for indicated times were fixed and stained for CSF-1R (anti-Cter sc-692 or anti-Nter sc-365719) or non-relevant antibodies (Rabbit or Mouse IgG respectively) and Dapi, followed by imaging flow cytometry (Amnis) and quantification of nuclear CSF-1R (n = 1, mean +/- SD, MFI : mean fluorescence intensity, A.U. : arbitrary units, ***P<0.001)

Issue # 5: If the authors could compare the same genes in their siRNA experiments between monocytes and macrophages as in Figure 4E and supplementary Figure 5E, this would help support their point that CSF-1R acts differently on genes depending on the cell's differentiation status.

Our answer: We thank the reviewer for this very relevant point. To follow this suggestion, we selected 2 genes. The first one is *KLF6* gene whose expression increases in monocytes when CSF-1R is down-regulated while its expression is not affected by CSF-1R siRNA in macrophages. The other one is *CSF2RB*, which is not significantly modified by CSF-1R down-regulation in monocytes while its expression is dramatically decreased by CSF-1R down-regulation in macrophages.

In the manuscript we added in blue: « siRNA mediated down-regulation of CSF-1R in CSF-1 treated monocytes induced a significant decrease in *YY1*, *ASXL1*, *CBL* and *CJUN* expression (Supplementary Fig. 7f), further enforcing that CSF-1R may directly or indirectly participate to the transcription of these genes in macrophages. We then selected to genes whose expression upon CSF-1R down-regulation was compared in monocytes and macrophages. The first one is *KLF6* whose expression increases in monocytes when CSF-1R is down-regulated (Fig. 4e) while being not affected by CSF-1R down-regulation in macrophages (Supplementary Fig. 7g). The other one is *CSF2RB*, which is not significantly modified by CSF-1R siRNA in monocytes (Fig. 4e) while its expression is dramatically decreased by CSF-1R down-regulation in macrophages (Supplementary Fig. 7g).

mRNA expression (qPCR) of *CFMS*, *KLF6* and *CSF2RB* genes in monocytes induced to differentiate into macrophages by exposure to 100 ng/mL CSF-1 and transfected with control or CSF-1R siRNA (mean +/- SEM of 4 or 3 independent experiments, normalization to control siRNA *p<0.05, **P<0.01, ns : non significant).

Reviewers' comments:

Reviewer #3 (Remarks to the Author):

The manuscript by Bencheikh, et al, provides evidence for the nuclear localization of full-length CSF-1R in monocytes and macrophages. Using ChIP-seq experiments, CSF-1R is shown to target H3K4me1-enriched non-promoter regions in monocytes interacting with EGR1 to decrease expression. In contrast, in macrophages CSF-1R targets H3K4me3-enriched promoter regions interacting with ELK and YY1 and associated with increased expression.

This manuscript was previously reviewed by two reviewers, one whose comments focused on the quality of the ChIP-seq experiments, and the other whose comments were largely focused on the evidence for CSF-1R nuclear localization. My expertise lies more with ChIP-seq experimentation and analysis, and thus I will comment primarily on this aspect of the paper and the previous reviewers' concerns. Overall, I agree with the previous reviewers that this work is of broad molecular and clinical research interest.

Even before reading the concerns from the first reviewer, I also was concerned by the quality of the ChIP-seq data. As the authors state, since CSF-1R does not appear to be directly binding DNA, but is indirectly associated with DNA through its interaction with other DNA-binding proteins, this may account for the weakness of the signal as I would expect that cross-linking would not be as effective as if there were direct interactions with DNA. But my concern initially arose based on the CSF-1R and EGR1 peaks being called based on an uncorrected p-value of 0.01, especially an average of 145K in macrophages. I assume that using a more accepted FDR (q-value) cut-off, such as was done for the histone ChIP-seq experiments, resulted in few to no peaks. Using the much more lenient uncorrected p-value resulted in peak counts that are not well-supported, especially as the authors say the amount of CSF-1R is likely low in the nucleus. Even though the authors use only the small fraction that are identified in all three samples, it is unclear whether these are reasonably strong in any single sample that would bolster confidence in their validity. The peak strength in individual replicates is not given, so this cannot be determined. As an alternative, I would urge the use of the IDR software package (<https://sites.google.com/site/anshulkundaje/projects/idr>) that specifically considers multiple replicates in the calling of reproducible peaks. The figures provided in the response suggest that there are at least some sites that have strong, reproducible signal, but the analysis is based on a few thousand peaks, not just these three. I would also suggest determining the correlations of the signals across the replicates to assess reproducibility.

Specific sites were tested with ChIP-qPCR, and the results seem pretty variable with some samples very close to fold enrichment = 1 and then some extremely high enrichments. My bet is that especially in the case of the C-terminal antibody, it was the same sample responsible for the extremely high enrichments which may indicate something strange in that sample. If IgG experiments were performed independently in each samples, it may indicate that the IgG level was incorrectly low, inflating the enrichments in these regions. It would also be good to see some negative controls – regions where you don't expect CSF-1R to be binding to provide a better comparison for these data.

The manuscript states that nearly all of the common peaks in monocytes were in intergenic and intronic regions, which is not surprising since nearly all of the genome is intergenic or intronic. A more interesting result would be whether the distribution of peaks is significantly different from the background distributions of these regions in the genome. This would make the distribution in macrophages truly significant, as 20% to promoters, 8% to exons, and 4.5% to 5'UTRs would represent a huge enrichment compared to the background.

Looking at the figures in Supp 7, though, I am a bit concerned by the diffuse nature of the CSF-1R signal. Normally, TF ChIP-seq peaks are rather punctate, with signal spanning just a couple hundred bases at most. Though not indicated clearly, the enriched signal in Supp 7d looks like it spans >1000 bases, on par with the H3K4me3 peaks that do normally span largest regions.

For PU.1, I'm not so sure that CSF-1R "moves" as much as it gains an additional binding site in macrophages. The signal in macrophages at the identified monocyte peak looks just as strong. In general, it is difficult to compare signals across experiments when the y-axes are all different. I know this is due to changes in sequencing depths, but then these signals should be normalized based on sequencing depth and presented with common y-axis units. It is very easy to show a signal in one experiment and not in a second simply by manipulating the y-axis.

Though I am not an expert at interpreting co-IP blots, these look convincing to me, as well as the EGR1 knockout experiments. Thus, I think the data establishing an interaction between these is robust. The effect on expression in monocytes is a bit oversold. I assume that all of the genes tested in by qPCR in Fig 4e were targets of CSF-1R. If yes, then of the seven, four show increase, two no change, and one a decrease in expression with the CSF-1R siRNA. Even with the PU.1 and LUCIFERASE results, this hardly warrants a general statement that CSF-1R negatively regulates genes in monocytes. An RNA-seq analysis of all target genes would be needed to make that type of general claim.

Panel 6F is not informative and is completely expected given that RNA-seq quantification is relative within a sample, not absolute. There is no way to compare the absolute expression level across all genes between two experiments.

In the CMML analysis, peaks are compared between single individuals (Fig 8d). Given the lack of overlap between even two normal individuals, this is not robust. For this comparison, it should require at least two normal and two CMML patients using reproducible peaks.

Overall, there is definitely strong evidence for some of the claims being made. But, without a more robust annotation of CSF-1R ChIP-seq peaks, it is difficult to confirm other claims being made. It may be beneficial to consult a computational genomics expert for advice on this.

Our response to reviewer #3

Remarks to the Author: *The manuscript by Bencheikh, et al, provides evidence for the nuclear localization of full-length CSF-1R in monocytes and macrophages. Using ChIP-seq experiments, CSF-1R is shown to target H3K4me1-enriched non-promoter regions in monocytes interacting with EGR1 to decrease expression. In contrast, in macrophages CSF-1R targets H3K4me3-enriched promoter regions interacting with ELK and YY1 and associated with increased expression. This manuscript was previously reviewed by two reviewers, one whose comments focused on the quality of the ChIP-seq experiments, and the other whose comments were largely focused on the evidence for CSF-1R nuclear localization. My expertise lies more with ChIP-seq experimentation and analysis, and thus I will comment primarily on this aspect of the paper and the previous reviewers' concerns. Overall, I agree with the previous reviewers that this work is of broad molecular and clinical research interest.*

Even before reading the concerns from the first reviewer, I also was concerned by the quality of the ChIP-seq data. As the authors state, since CSF-1R does not appear to be directly binding DNA, but is indirectly associated with DNA through its interaction with other DNA-binding proteins, this may account for the weakness of the signal as I would expect that cross-linking would not be as effective as if there were direct interactions with DNA. But my concern initially arose based on the CSF-1R and EGR1 peaks being called based on an uncorrected p-value of 0.01, especially an average of 145K in macrophages. I assume that using a more accepted FDR (q-value) cut-off, such as was done for the histone ChIP-seq experiments, resulted in few to no peaks. Using the much more lenient uncorrected p-value resulted in peak counts that are not well-supported, especially as the authors say the amount of CSF-1R is likely low in the nucleus.

Even though the authors use only the small fraction that are identified in all three samples, it is unclear whether these are reasonably strong in any single sample that would bolster confidence in their validity. The peak strength in individual replicates is not given, so this cannot be determined. As an alternative, I would urge the use of the IDR software package (<https://sites.google.com/site/anshulkundaje/projects/idr>) that specifically considers multiple replicates in the calling of reproducible peaks. The figures provided in the response suggest that there are at least some sites that have strong, reproducible signal, but the analysis is based on a few thousand peaks, not just these three. I would also suggest determining the correlations of the signals across the replicates to assess reproducibility.

Our response: We are very grateful to the reviewer for these comments and useful advices. We have re-analyzed all our previous experiments following the methods suggested by the reviewer, with the help of a computational genomics expert (Camille Lobry).

First, we apologize for the typo error in the previous version of the manuscript as CSF-1R and EGR1 peaks had been called based on an uncorrected P-value of 0.001, not 0.01.

To re-analyze all our ChIP-seq data, we used MACS2 algorithm, selecting the broad option given the profile of CSF1R binding (more details are given below), not building the shifting model (using extsize, whose choice was based on our library size) and using the default q-value cut off (all the detailed parameters used for the analysis are provided in the revised Material and Method).

This method allowed identification of 33,324, 37,924 and 39,402 peaks respectively in the 3 CSF1R ChIP-Seq replicates performed in healthy donor monocyte samples, with an overlap of 4,980 peaks. Following the reviewer's advice, we performed IDR analysis on these peaks using ENCODE3 guidelines (using replicates, pooled replicate, pseudo-replicates and pooled pseudo-replicates), which identified an optimal set of 3,054 peaks. The overlap between common peaks in the 3 replicates and IDR optimal set was 2,303 peaks, including all the peaks subsequently characterized in our studies.

Script used for IDR

```
#!/bin/bash

REP1=[name of replicate #1]
REP2=[name of replicate #2]
REP3=[name of replicate #3]
REP1_IP_BAM_FILE=[file name of replicate #1 IP BAM alignment]
REP2_IP_BAM_FILE=[file name of replicate #2 IP BAM alignment]
REP3_IP_BAM_FILE=[file name of replicate #3 IP BAM alignment]
REP1_Input_BAM_FILE=[file name of replicate #1 input BAM alignment]
REP2_Input_BAM_FILE=[file name of replicate #2 input BAM alignment]
REP3_Input_BAM_FILE=[file name of replicate #3 input BAM alignment]

# =====
# Split BAM files to generate
# pseudo replicates.
# =====
split_bam.py -b ${REP1_IP_BAM_FILE} -o1 "${REP1}"_PR1.bam -o2 "${REP1}"_PR2.bam -p
0.5
split_bam.py -b ${REP2_IP_BAM_FILE} -o1 "${REP2}"_PR1.bam -o2 "${REP2}"_PR2.bam -p
0.5
split_bam.py -b ${REP3_IP_BAM_FILE} -o1 "${REP3}"_PR1.bam -o2 "${REP3}"_PR2.bam -p
0.5

# =====
# Perform MACS2 peakcalling.
# =====
# =====
# On true replicates.
# =====
macs2 callpeak -t ${REP1_IP_BAM_FILE} -c ${REP1_Input_BAM_FILE} -g hs -n
MACS2 "${REP1}"_BROAD -f BAM --nomodel --broad &
macs2 callpeak -t ${REP2_IP_BAM_FILE} -c ${REP2_Input_BAM_FILE} -g hs -n
MACS2 "${REP2}"_BROAD -f BAM --nomodel --broad &
macs2 callpeak -t ${REP3_IP_BAM_FILE} -c ${REP3_Input_BAM_FILE} -g hs -n
MACS2 "${REP3}"_BROAD -f BAM --nomodel --broad &&

wait

# =====
# On pooled true replicates.
# =====
macs2 callpeak -t ${REP1_IP_BAM_FILE} ${REP2_IP_BAM_FILE} ${REP3_IP_BAM_FILE} -c
${REP1_Input_BAM_FILE} ${REP2_Input_BAM_FILE} ${REP3_Input_BAM_FILE} -g hs -n
MACS2_POOLED_BROAD -f BAM --nomodel --broad

# =====
# On pseudo replicates.
# =====
macs2 callpeak -t "${REP1}"_PR1.bam -c ${REP1_Input_BAM_FILE} -g hs -n
MACS2 "${REP1}"_BROAD_pr1 -f BAM --nomodel --broad &
macs2 callpeak -t "${REP1}"_PR2.bam -c ${REP1_Input_BAM_FILE} -g hs -n
MACS2 "${REP1}"_BROAD_pr2 -f BAM --nomodel --broad &

macs2 callpeak -t "${REP2}"_PR1.bam -c ${REP2_Input_BAM_FILE} -g hs -n
MACS2 "${REP2}"_BROAD_pr1 -f BAM --nomodel --broad &
```

```

macs2 callpeak -t "${REP2}"_PR2.bam -c ${REP2_Input_BAM_FILE} -g hs -n
MACS2 "${REP2}"_BROAD_pr2 -f BAM --nomodel --broad &

macs2 callpeak -t "${REP3}"_PR1.bam -c ${REP3_Input_BAM_FILE} -g hs -n
MACS2 "${REP3}"_BROAD_pr1 -f BAM --nomodel --broad &
macs2 callpeak -t "${REP3}"_PR2.bam -c ${REP3_Input_BAM_FILE} -g hs -n
MACS2 "${REP3}"_BROAD_pr2 -f BAM --nomodel --broad &&

wait

# =====
# On pooled pseudo replicates.
# =====
macs2 callpeak -t "${REP1}"_PR1.bam "${REP2}"_PR1.bam "${REP3}"_PR1.bam -c
${REP1_Input_BAM_FILE} ${REP2_Input_BAM_FILE} ${REP3_Input_BAM_FILE} -g hs -n
MACS2_BROAD_PPR1 -f BAM --nomodel --broad &
macs2 callpeak -t "${REP1}"_PR2.bam "${REP2}"_PR2.bam "${REP3}"_PR2.bam -c
${REP1_Input_BAM_FILE} ${REP2_Input_BAM_FILE} ${REP3_Input_BAM_FILE} -g hs -n
MACS2_BROAD_PPR2 -f BAM --nomodel --broad &&

wait

gzip -k *broadPeak

#source activate idr_env

REP1_PEAK_FILE="MACS2_${REP1}"_BROAD_peaks.broadPeak"
REP2_PEAK_FILE="MACS2_${REP2}"_BROAD_peaks.broadPeak"
REP3_PEAK_FILE="MACS2_${REP3}"_BROAD_peaks.broadPeak"
REP1_PR1_PEAK_FILE="MACS2_${REP1}"_BROAD_pr1_peaks.broadPeak"
REP1_PR2_PEAK_FILE="MACS2_${REP1}"_BROAD_pr2_peaks.broadPeak"
REP2_PR1_PEAK_FILE="MACS2_${REP2}"_BROAD_pr1_peaks.broadPeak"
REP2_PR2_PEAK_FILE="MACS2_${REP2}"_BROAD_pr2_peaks.broadPeak"
REP3_PR1_PEAK_FILE="MACS2_${REP3}"_BROAD_pr1_peaks.broadPeak"
REP3_PR2_PEAK_FILE="MACS2_${REP3}"_BROAD_pr2_peaks.broadPeak"
PPR1_PEAK_FILE="MACS2_BROAD_PPR1_peaks.broadPeak"
PPR2_PEAK_FILE="MACS2_BROAD_PPR2_peaks.broadPeak"
POOLED_PEAK_FILE="MACS2_POOLED_BROAD_peaks.broadPeak"
BLACKLIST="wgEncodeDacMapabilityConsensusExcludable.bed.gz"
IDR_THRESH=0.05

# =====
# Perform IDR analysis.
# Generate a plot and IDR output with additional columns including IDR scores.
# =====
idr --samples ${REP1_PEAK_FILE} ${REP2_PEAK_FILE} --peak-list ${POOLED_PEAK_FILE} -
-input-file-type broadPeak --output-file IDR "${REP1}"_VS_"${REP2}" --rank
signal.value --soft-idr-threshold ${IDR_THRESH} --plot --use-best-multisummit-IDR
2>IDR.out
idr --samples ${REP1_PEAK_FILE} ${REP3_PEAK_FILE} --peak-list ${POOLED_PEAK_FILE} -
-input-file-type broadPeak --output-file IDR "${REP1}"_VS_"${REP3}" --rank
signal.value --soft-idr-threshold ${IDR_THRESH} --plot --use-best-multisummit-IDR
2>>IDR.out
idr --samples ${REP2_PEAK_FILE} ${REP3_PEAK_FILE} --peak-list ${POOLED_PEAK_FILE} -
-input-file-type broadPeak --output-file IDR "${REP2}"_VS_"${REP3}" --rank
signal.value --soft-idr-threshold ${IDR_THRESH} --plot --use-best-multisummit-IDR
2>>IDR.out

idr --samples ${REP1_PR1_PEAK_FILE} ${REP1_PR2_PEAK_FILE} --peak-list
${REP1_PEAK_FILE} --input-file-type broadPeak --output-file IDR "${REP1}"_PR --rank
signal.value --soft-idr-threshold ${IDR_THRESH} --plot --use-best-multisummit-IDR
2>>IDR.out
idr --samples ${REP2_PR1_PEAK_FILE} ${REP2_PR2_PEAK_FILE} --peak-list
${REP2_PEAK_FILE} --input-file-type broadPeak --output-file IDR "${REP2}"_PR --rank
signal.value --soft-idr-threshold ${IDR_THRESH} --plot --use-best-multisummit-IDR
2>>IDR.out
idr --samples ${REP3_PR1_PEAK_FILE} ${REP3_PR2_PEAK_FILE} --peak-list
${REP3_PEAK_FILE} --input-file-type broadPeak --output-file IDR "${REP3}"_PR --rank

```

```

signal.value --soft-idr-threshold ${IDR_THRESH} --plot --use-best-multisummit-IDR
2>>IDR.out

idr --samples ${PPR1_PEAK_FILE} ${PPR2_PEAK_FILE} --peak-list ${POOLED_PEAK_FILE} -
-input-file-type broadPeak --output-file IDR_PPR --rank signal.value --soft-idr-
threshold ${IDR_THRESH} --plot --use-best-multisummit-IDR 2>>IDR.out

# =====
# Get peaks passing IDR threshold of 5%
# =====
IDR_THRESH_TRANSFORMED=$(awk -v p=${IDR_THRESH} 'BEGIN{print -log(p)/log(10)}')

awk 'BEGIN{OFS="\t"} $11>=${IDR_THRESH_TRANSFORMED}' {print
$1,$2,$3,$4,$5,$6,$7,$8,$9,$10}' IDR_${REP1}_VS_${REP2}" | sort | uniq | sort -
k7n,7n | gzip -nc > "${REP1}_VS_${REP2}.IDR0.05.broadPeak.gz
NPEAKS_IDR1=$(zcat "${REP1}_VS_${REP2}.IDR0.05.broadPeak.gz | wc -l)

awk 'BEGIN{OFS="\t"} $11>=${IDR_THRESH_TRANSFORMED}' {print
$1,$2,$3,$4,$5,$6,$7,$8,$9,$10}' IDR_${REP1}_VS_${REP3}" | sort | uniq | sort -
k7n,7n | gzip -nc > "${REP1}_VS_${REP3}.IDR0.05.broadPeak.gz
NPEAKS_IDR2=$(zcat "${REP1}_VS_${REP3}.IDR0.05.broadPeak.gz | wc -l)

awk 'BEGIN{OFS="\t"} $11>=${IDR_THRESH_TRANSFORMED}' {print
$1,$2,$3,$4,$5,$6,$7,$8,$9,$10}' IDR_${REP2}_VS_${REP3}" | sort | uniq | sort -
k7n,7n | gzip -nc > "${REP2}_VS_${REP3}.IDR0.05.broadPeak.gz
NPEAKS_IDR3=$(zcat "${REP2}_VS_${REP3}.IDR0.05.broadPeak.gz | wc -l)

awk 'BEGIN{OFS="\t"} $11>=${IDR_THRESH_TRANSFORMED}' {print
$1,$2,$3,$4,$5,$6,$7,$8,$9,$10}' IDR_${REP1}_PR | sort | uniq | sort -k7n,7n |
gzip -nc > "${REP1}_PR.IDR0.05.broadPeak.gz
NPEAKS_IDR4=$(zcat "${REP1}_PR.IDR0.05.broadPeak.gz | wc -l)

awk 'BEGIN{OFS="\t"} $11>=${IDR_THRESH_TRANSFORMED}' {print
$1,$2,$3,$4,$5,$6,$7,$8,$9,$10}' IDR_${REP2}_PR | sort | uniq | sort -k7n,7n |
gzip -nc > "${REP2}_PR.IDR0.05.broadPeak.gz
NPEAKS_IDR5=$(zcat "${REP2}_PR.IDR0.05.broadPeak.gz | wc -l)

awk 'BEGIN{OFS="\t"} $11>=${IDR_THRESH_TRANSFORMED}' {print
$1,$2,$3,$4,$5,$6,$7,$8,$9,$10}' IDR_${REP3}_PR | sort | uniq | sort -k7n,7n |
gzip -nc > "${REP3}_PR.IDR0.05.broadPeak.gz
NPEAKS_IDR6=$(zcat "${REP3}_PR.IDR0.05.broadPeak.gz | wc -l)

awk 'BEGIN{OFS="\t"} $11>=${IDR_THRESH_TRANSFORMED}' {print
$1,$2,$3,$4,$5,$6,$7,$8,$9,$10}' IDR_PPR | sort | uniq | sort -k7n,7n | gzip -nc >
PPR.IDR0.05.broadPeak.gz
NPEAKS_IDR7=$(zcat PPR.IDR0.05.broadPeak.gz | wc -l)

# =====
# Filter using black list
# =====
bedtools intersect -v -a "${REP1}_VS_${REP2}.IDR0.05.broadPeak.gz -b
${BLACKLIST} | grep -P 'chr[\dXY]+[\t]' | awk 'BEGIN{OFS="\t"} {if ($5>1000)
$5=1000; print $0}' | gzip -nc > "${REP1}_VS_${REP2}.IDR0.05.filt.broadPeak.gz
NPEAKS_IDR1_filt=$(zcat "${REP1}_VS_${REP2}.IDR0.05.filt.broadPeak.gz | wc -l)

bedtools intersect -v -a "${REP1}_VS_${REP3}.IDR0.05.broadPeak.gz -b
${BLACKLIST} | grep -P 'chr[\dXY]+[\t]' | awk 'BEGIN{OFS="\t"} {if ($5>1000)
$5=1000; print $0}' | gzip -nc > "${REP1}_VS_${REP3}.IDR0.05.filt.broadPeak.gz
NPEAKS_IDR2_filt=$(zcat "${REP1}_VS_${REP3}.IDR0.05.filt.broadPeak.gz | wc -l)

bedtools intersect -v -a "${REP2}_VS_${REP3}.IDR0.05.broadPeak.gz -b
${BLACKLIST} | grep -P 'chr[\dXY]+[\t]' | awk 'BEGIN{OFS="\t"} {if ($5>1000)
$5=1000; print $0}' | gzip -nc > "${REP2}_VS_${REP3}.IDR0.05.filt.broadPeak.gz
NPEAKS_IDR3_filt=$(zcat "${REP2}_VS_${REP3}.IDR0.05.filt.broadPeak.gz | wc -l)

bedtools intersect -v -a "${REP1}_PR.IDR0.05.broadPeak.gz -b ${BLACKLIST} | grep -
P 'chr[\dXY]+[\t]' | awk 'BEGIN{OFS="\t"} {if ($5>1000) $5=1000; print $0}' | gzip
-nc > "${REP1}_PR.IDR0.05.filt.broadPeak.gz

```

```

NPEAKS_IDR4_filt=$(zcat "${REP1}"_PR.IDR0.05.filt.broadPeak.gz | wc -l)

bedtools intersect -v -a "${REP2}"_PR.IDR0.05.broadPeak.gz -b ${BLACKLIST} | grep -
P 'chr[\dXY]+[\ \t]' | awk 'BEGIN{OFS="\t"} {if ($5>1000) $5=1000; print $0}' | gzip
-nc > "${REP2}"_PR.IDR0.05.filt.broadPeak.gz
NPEAKS_IDR5_filt=$(zcat "${REP2}"_PR.IDR0.05.filt.broadPeak.gz | wc -l)

bedtools intersect -v -a "${REP3}"_PR.IDR0.05.broadPeak.gz -b ${BLACKLIST} | grep -
P 'chr[\dXY]+[\ \t]' | awk 'BEGIN{OFS="\t"} {if ($5>1000) $5=1000; print $0}' | gzip
-nc > "${REP3}"_PR.IDR0.05.filt.broadPeak.gz
NPEAKS_IDR6_filt=$(zcat "${REP3}"_PR.IDR0.05.filt.broadPeak.gz | wc -l)

bedtools intersect -v -a PPR.IDR0.05.broadPeak.gz -b ${BLACKLIST} | grep -P
'chr[\dXY]+[\ \t]' | awk 'BEGIN{OFS="\t"} {if ($5>1000) $5=1000; print $0}' | gzip -
nc > PPR.IDR0.05.filt.broadPeak.gz
NPEAKS_IDR7_filt=$(zcat PPR.IDR0.05.filt.broadPeak.gz | wc -l)

```

Changes in the manuscript: We have now indicated in the figure the MACS2 common CSF-1R peaks in untreated monocytes (day 0) and in monocytes treated with CSF1 for 3 days (day 3) by showing bed files. All tag density representations have been normalized to the depth of sequencing and compared tracks are now displayed at a similar y-axis value. Implementation of all the figures using these normalized Bigwigs strongly improves the quality of CSF-1R peaks (new Figures 3, 4, 8, 9, and new supplemental Figures 3, 4, 5 and 6). To illustrate the co-localization of CSF-1R peaks with H3K4me1 and EGR1 peaks in untreated monocytes (day 0) and with H3K4me3 peaks in monocytes treated for 3 days with CSF-1 (day 3), we now provide ranking heatmaps on these peaks and on TSS as illustrated below

We also followed the suggestion to measure Pearson correlation between replicates, which argued for the reproducibility of ChIPseq experiments. Correlation scatters of CSF-1R peaks in untreated monocytes (day 0, left panel) and CSF-1-treated monocytes (day 3, right) are shown below:

All the subsequent analyses have been performed with this new set of peaks, including motif discovery, co-occupancy with EGR1, H3K4me1 or H3K4me3 and Gene Ontology. Importantly, the results are validated and enforced our original conclusions. Again, we would like to thank the reviewer for all these insightful advices

Remarks to the Author: *Specific sites were tested with ChIP-qPCR, and the results seem pretty variable with some samples very close to fold enrichment = 1 and then some extremely high enrichments. My bet is that especially in the case of the C-terminal antibody, it was the same sample responsible for the extremely high enrichments which may indicate something strange in that sample. If IgG experiments were performed independently in each sample, it may indicate that the IgG level was incorrectly low, inflating the enrichments in these regions. It would also be good to see some negative controls – regions where you don't expect CSF-1R to be binding to provide a better comparison for these data.*

Our response: The reviewer was right in detecting this abnormal sample. Indeed, we had initially missed the fact that, with some IgG, Ct values were abnormally high, generating an artificially high enrichment. We have repeated the experiments (ChIP-qPCR) in untreated monocytes with two CSF-1R antibodies. We have included the first intron of *RAC2* gene as a negative control (CSF-1R does not bind this region) (see figure S3b). These experiments were also repeated in monocytes treated with CSF-1 for 3 days, with some modifications in target genes as *IDH1*, which had been tested initially, was no more part of our optimal set of annotated genes and we included two new targets (*YY1* and *IL6R*) (see Figure S6c).

Changes in the manuscript: Results of the new ChIP-qPCR experiments, including a control target gene named *RAC2* used as a normalizer, are shown on supplemental Figure 3b and 6b.

Remarks to the Author: *The manuscript states that nearly all of the common peaks in monocytes were in intergenic and intronic regions, which is not surprising since nearly all of the genome is intergenic or intronic. A more interesting result would be whether the distribution of peaks is significantly different from the background distributions of these regions in the genome. This would make the distribution in macrophages truly significant, as 20% to promoters, 8% to exons, and 4.5% to 5'UTRs would represent a huge enrichment compared to the background.*

Our response: Thank you for this important suggestion. We have now used CEAS analysis (<http://liulab.dfci.harvard.edu/CEAS/>) to determine the fractions of peaks in different parts of the genome and compared it to the background distribution of the different regions in a reference genome. In untreated monocytes, we observed that CSF-1R peaks were specifically enriched in intergenic regions. In monocytes treated with CSF-1 for 3 days, CSF-

1R peaks were enriched at promoter-TSS, consistent with their colocalization with H3K4me3 peaks.

Changes in the manuscript: The background distribution of the reference genome regions is depicted on Figure 3b and the impact of using this reference genome on data interpretation led to modifications all along the revised manuscript.

Remarks to the Author: *Looking at the figures in Supp 7, though, I am a bit concerned by the diffuse nature of the CSF-1R signal. Normally, TF ChIP-seq peaks are rather punctate, with signal spanning just a couple hundred bases at most. Though not indicated clearly, the enriched signal in Supp 7d looks like it spans >1000 bases, on par with the H3K4me3 peaks that do normally span largest regions.*

Our response: As noticed by the reviewer, CSF-1R peaks cover relatively broad regions. This pattern of binding is still observed after having applied all the corrections previously suggested by the reviewer. A similar pattern has been described for other receptors re-localizing in the nucleus such as EGFR (Mikula M *et al.*, Nucleic Acids Research, 44:10150-10164, 2016) and may be explained by the fact that these receptors are not transcription factors per se and bind DNA through interacting with other proteins.

Changes in the manuscript: We have used the broad peak calling option of MACS2 algorithm all along the study to deal with the pattern of CSF1R peaks

Remarks to the Author: *For PU.1, I'm not so sure that CSF-1R "moves" as much as it gains an additional binding site in macrophages. The signal in macrophages at the identified monocyte peak looks just as strong. In general, it is difficult to compare signals across experiments when the y-axes are all different. I know this is due to changes in sequencing depths, but then these signals should be normalized based on sequencing depth and presented with common y-axis units. It is very easy to show a signal in one experiment and not in a second simply by manipulating the y-axis.*

Our response: Again, we can only agree with this comment and have followed the reviewer's advices.

Changes in the manuscript: All tag density representations have now been normalized to the depth of sequencing and compared tracks are now displayed at a similar y-axis value. Regarding *PU.1* gene, Figure 3c shows a peak at the promoter level and another peak in the last intron of untreated monocytes. After 3 days of exposure to CSF-1, a unique peak is detected in the first intron, close to the TSS, which is shown as supplemental Figure 6e.

Remarks to the Author: *Though I am not an expert at interpreting co-IP blots, these look convincing to me, as well as the EGR1 knockout experiments. Thus, I think the data establishing an interaction between these is robust. The effect on expression in monocytes is a bit oversold. I assume that all of the genes tested in by qPCR in Fig 4e were targets of CSF-1R. If yes, then of the seven, four show increase, two no change, and one a decrease in expression with the CSF-1R siRNA. Even with the PU.1 and LUCIFERASE results, this hardly warrants a general statement that CSF-1R negatively regulates genes in monocytes. An RNA-seq analysis of all target genes would be needed to make that type of general claim.*

Our response: We totally agree with the reviewer that the conclusion that CSF-1R was globally a negative regulator of gene expression in untreated monocytes was an over interpretation of our data.

Modifications: We have removed from the manuscript, including in the abstract, the statement that CSF-1R was a negative regulator of gene expression in untreated monocytes and became a positive regulator in CSF-1 treated cells.

Remarks to the Author: *Panel 6F is not informative and is completely expected given that RNA-seq quantification is relative within a sample, not absolute. There is no way to compare the absolute expression level across all genes between two experiments.*

Our response: We agree with the reviewer that Panel 6F in the previous version was not informative.

Changes in the manuscript: The panel has been removed from the new version of the manuscript as this comment did not bring much to the manuscript.

Remarks to the Author: *In the CMML analysis, peaks are compared between single individuals (Fig 8d). Given the lack of overlap between even two normal individuals, this is not robust. For this comparison, it should require at least two normal and two CMML patients using reproducible peaks.*

Our response: We have completely changed our analysis of patient data. We have performed more ChIP-Seq analyses, of which the majority cannot be included as the decreased expression of CSF-1R in the patient cells precluded a correct immunoprecipitation. Of note, we had used initially a rabbit polyclonal anti-CSF-1R C-terminal antibody (sc-692) to increase our chance to immunoprecipitate CSF-1R when expressed at low level in patient cells. This antibody is no more produced by SantaCruz. The new ChIP-Seq experiments were done with a monoclonal antibody targeting the C-terminus of the protein (sc-46662), which is less efficient. The new ChIP-Seq experiments were done with a monoclonal antibody targeting the C-terminus of the protein (sc-46662). We have generated a bed file with reproducible peaks by intersecting the 4,980 peaks obtained with the 3 donors using anti-CSF-1R monoclonal N-terminal antibody (sc-46662) with the peaks obtained in a donor with the anti-CSF-1R C-terminal antibody (sc-692) and we obtained 2,276 common peaks that are robust. To illustrate how heterogeneous the patients cells may be, starting with these 2,276 common peaks in healthy donors monocytes, 1,800 (79%) were still present in one of the patient samples while only 716 (31%) were found in another one. Although we have screened up to 60 patient samples by immunoblotting for CSF-1R, we cannot provide a large scale analysis of ChIP-Seq data. This is why we have chosen to show that some peaks appear to be conserved in the analyzed samples whereas other are lost and could hardly provide a more detailed conclusion.

Changes in the manuscript: "In those in which we succeeded in carrying out CSF-1R immunoprecipitation, we generated a bed file of 2,276 reproducible peaks (intersection of 4,980 peaks obtained with the monoclonal N-terminal CSF-1R antibody with 23,480 peaks obtained with the polyclonal, C-terminal CSF-1R antibody). Some enrichments on chromatin sites were conserved in healthy donor and patient monocytes, *e.g* on intergenic area on chromosome 5 and *CEBPA* and *BCL2* genes (Fig. 9c) whereas others were lost in some patient cells, *e.g.* in *BPIFB1* gene, downstream of *AP1M1* gene and in intergenic areas on chromosome 11 and 22 (Fig. 9d). These results indicated a heterogeneous disruption of CSF-1R interaction with chromatin in CMML patient monocytes."

Remarks to the Author: *Overall, there is definitely strong evidence for some of the claims being made. But, without a more robust annotation of CSF-R1 ChIP-seq peaks, it is difficult to*

confirm other claims being made. It may be beneficial to consult a computational genomics expert for advice on this.

Our response: We have been re-analyzing our whole set of data with a computational genomics expert (Camille Lobry) who generated algorithms to better perform these analyses, which greatly improved our interpretation of the generated data. Importantly, the global message remains true, but detailed analysis improves the interpretation of all our data.

Reviewer #3 (Remarks to the Author):

I believe the authors have addressed my previous questions very well, and I commend them on their thoroughness in re-analyzing the genomic data. I have much more confidence in the accuracy and robustness of the new annotations.

I do believe that the manuscript could be further improved through a careful editing by a native English speaker. There are several places where the grammar is not quite right. Some of the phrasing of different observed phenomena could be improved as well. For instance, upon differentiation from monocytes to macrophages, I do not believe that CSF-1R "moves" along chromatin. As it likely does not bind directly to DNA but rather localizes with DNA through interactions with other factors, the changes seen likely reflect either a dissociation with one factor and re-association with another that then localizes to alternative genomic locations, or completely new nuclear CSF-1R binds to different factors in macrophages and previously bound CSF-1R are degraded during differentiation. Either way, they don't "move", which implies there is some type of translocation of the same CSF-1R complex. I am not overly concerned with these issues and believe they can be readily resolved.

Terry Furey

Reviewer #3 (Remarks to the Author):

I believe the authors have addressed my previous questions very well, and I commend them on their thoroughness in re-analyzing the genomic data. I have much more confidence in the accuracy and robustness of the new annotations.

I do believe that the manuscript could be further improved through a careful editing by a native English speaker. There are several places where the grammar is not quite right. Some of the phrasing of different observed phenomena could be improved as well. For instance, upon differentiation from monocytes to macrophages, I do not believe that CSF-1R "moves" along chromatin. As it likely does not bind directly to DNA but rather localizes with DNA through interactions with other factors, the changes seen likely reflect either a dissociation with one factor and re-association with another that then localizes to alternative genomic locations, or completely new nuclear CSF-1R binds to different factors in macrophages and previously bound CSF-1R are degraded during differentiation. Either way, they don't "move", which implies there is some type of translocation of the same CSF-1R complex. I am not overly concerned with these issues and believe they can be readily resolved.

Terry Furey

Our response to reviewer #3

We would like to thank the reviewer for noticing this and we do agree that CSF-1R does not move along chromatin but instead is relocalized through different partner interaction. As suggested, corresponding changes have been made in the text :

Introduction

Paragraph_4

« Upon CSF-1 exposure, which induces monocyte differentiation into macrophages, CSF-1R **localization** on chromatin **changes** within a few hours ; **it** colocalizes with H3K4me3, and promotes gene expression through interaction with transcription factors YY1 and ELK.»

Results

« Ranking heatmaps centered on TSS indicated that CSF-1R **is relocalized** around the TSS where it colocalizes with H3K4me3 in macrophages (Fig. 7b), as exemplified for MAFB, JUN and MYC genes (Supplementary Figure 6D). »

« CSF-1R **location** also **changed** from the promoter region of PU.1 gene in monocytes to PU.1 first intron in macrophages where it colocalizes with H3K4me3 mark (Supplementary Figure 6E).»

Discussion

« Upon CSF-1-induced monocyte differentiation into macrophage, nuclear CSF-1R **is relocated on** chromatin within a few hours to differentially influence gene expression through interaction with YY1 and ELK transcription factors.»